# VISUAL JIGSAW POST-TRAINING IMPROVES MLLMS

**Penghao Wu**[1]    **Yushan Zhang**[2]    **Haiwen Diao**[1]    **Bo Li**[1]    **Lewei Lu**[3]    **Ziwei Liu**[1]

[1]S-Lab, Nanyang Technological University
[2]Linköping University
[3]SenseTime Research

Project Page: `https://penghao-wu.github.io/visual_jigsaw/`

## ABSTRACT

Reinforcement learning based post-training has recently emerged as a powerful paradigm for enhancing the alignment and reasoning capabilities of multimodal large language models (MLLMs). While *vision-centric* post-training is crucial for enhancing MLLMs' intrinsic understanding of visual signals, current post-training paradigms are predominantly *text-centric*, where dense visual inputs are only leveraged to extract sparse cues for text-based reasoning. There exist a few approaches in this direction, however, they often still rely on text as an intermediate mediator or introduce additional visual generative designs. In this work, we introduce **Visual Jigsaw**, a generic *self-supervised* post-training framework designed to strengthen visual understanding in MLLMs. Visual Jigsaw is formulated as a general ordering task: visual inputs are partitioned, shuffled, and the model must reconstruct the visual information by producing the correct permutation in natural language. This naturally aligns with reinforcement learning from verifiable rewards (RLVR), requires no additional visual generative components, and derives its supervisory signal automatically without any annotations. We instantiate Visual Jigsaw across three visual modalities, including images, videos, and 3D data. Extensive experiments demonstrate substantial improvements in fine-grained perception, temporal reasoning, and 3D spatial understanding. Our findings highlight the potential of self-supervised vision-centric tasks in post-training MLLMs and aim to inspire further research on vision-centric pretext designs.

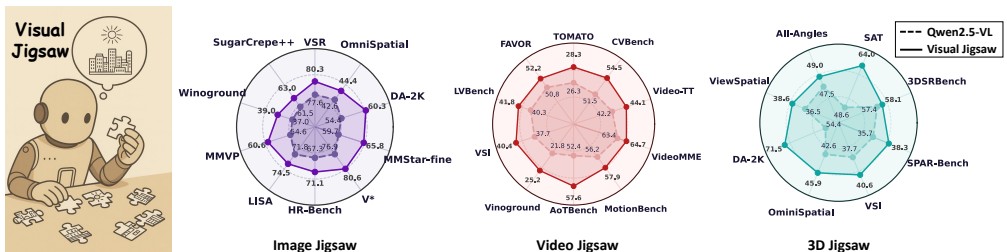

Figure 1: We propose **Visual Jigsaw**, a self-supervised post-training task that enhances visual perception and understanding in MLLMs. Training on visual jigsaw tasks substantially strengthens fine-grained perception, monocular spatial perception, and compositional visual understanding in images; temporal understanding in videos; and geometry-aware understanding in 3D, demonstrating its generality and effectiveness across modalities. For clearer visualization, the value ranges differ across benchmarks in each radar chart.

## 1 INTRODUCTION

Multimodal large language models (MLLMs) have recently demonstrated remarkable progress, achieving strong performance on a wide range of vision–language tasks. Following the success of Reinforcement Learning from Verifiable Reward (RLVR) (Lambert et al., 2024; Guo et al., 2025) in the large language models domain, which has unlocked substantial breakthroughs in complex reasoning abilities, the research community has largely shifted its focus toward replicating this success

in the multimodal domain. This has led to a predominant focus on advancing text-based Chain-of-Thought (CoT) multimodal reasoning to enhance multimodal mathematical and scientific reasoning (Huang et al., 2025; Meng et al., 2025; Yuan et al., 2025a; Wang et al., 2025f).

Within this paradigm, dense visual information often serves merely as contextual evidence, from which the model extracts sparse information to support text-based reasoning. Consequently, a deep, fine-grained understanding of the visual signal itself has been considerably undervalued. Some recent studies (Wang et al., 2025b;a) have shown that explicitly incorporating visual reconstruction objectives during the training of MLLMs can improve visual understanding. However, such approaches necessitate the integration of additional visual generation components and learning objectives onto the existing understanding-based MLLM architectures. Furthermore, it remains an open question whether forcing models to achieve pixel-level reconstruction is the optimal strategy for enhancing MLLMs' visual understanding. This raises a pivotal question: Can we enhance an MLLM's visual understanding without altering its architecture or output format?

Delving into the history of self-supervised visual representation learning reveals a rich set of pretext tasks, such as reconstruction-based approaches (He et al., 2022a) and discriminative approaches (He et al., 2020). In parallel, jigsaw-style tasks have emerged as lightweight yet effective paradigms: reordering shuffled image patches (Noroozi & Favaro, 2016), recovering video frame order (Ahsan et al., 2019). While these jigsaw-style approaches provide structural ordering signals, they have generally shown weaker performance compared to more dominant approaches and thus have not become mainstream in vision representation learning. Nevertheless, they demonstrate that the structural ordering jigsaw task, which can be viewed as a simpler version of the reconstruction/generation task, can still offer effective self-supervised signals without requiring pixel-level fidelity.

In this work, we introduce **Visual Jigsaw**, a self-supervised task designed for the RL post-training phase of MLLMs to enhance their visual perception and understanding. The task is formulated as a lightweight ordering problem: visual inputs are partitioned, permuted, and presented to the MLLM, which must then generate the correct permutation order using natural language. Importantly, this formulation requires no additional visual generative designs and is seamlessly compatible with existing MLLMs that produce text-only outputs. Moreover, this task naturally fits in the RLVR framework with deterministic ground-truth and requires no other annotations. We position Visual Jigsaw in the post-training phase, as solving it requires the model to already possess a foundational level of visual understanding. Furthermore, post-training with RL has been shown to offer stronger generalization than Supervised Fine-Tuning (SFT) (Huan et al., 2025; Chu et al., 2025; Chen et al., 2025a), enabling the model to better transfer the vision-centric skills acquired from the jigsaw task to downstream applications.

We implement Visual Jigsaw across three visual modalities: images, videos, and 3D data. Through a post-training phase with Group Relative Policy Optimization (GRPO) (Shao et al., 2024) on visual jigsaw tasks, we substantially improve the ability of MLLMs to perceive and comprehend these visual modalities (shown in Fig 1). In the image domain, we partition the input into patches, shuffle them, and require the model to recover the correct spatial arrangement. We find that this task enhances fine-grained perception, monocular spatial understanding, and compositional visual understanding. For video, we segment the input along the temporal axis, shuffle the clips, and challenge the model to reconstruct the original sequence, leading to marked improvements in temporal understanding. In the 3D domain, we sample points with distinct depth values from an RGB-D image, shuffle and annotate them in the RGB view, and require the model to recover their order from nearest to farthest, thereby augmenting its 3D perceptual capabilities.

Our main contributions are: **1)** We introduce Visual Jigsaw, a lightweight and verifiable self-supervised post-training task that enhances vision-centric perception and understanding capabilities in MLLMs. It requires no additional generative modules and integrates seamlessly with existing text-only models. **2)** We instantiate Visual Jigsaw across three visual modalities—images, videos, and 3D data—and demonstrate consistent improvements in fine-grained perception, temporal understanding, and 3D spatial reasoning, thereby establishing its generality and effectiveness. **3)** We highlight the potential of self-supervised tasks focused explicitly on the visual signal as a promising, complementary direction for enhancing the vision-centric abilities of MLLMs.

## 2 RELATED WORKS

### 2.1 SELF-SUPERVISED LEARNING

Self-supervised learning (SSL), wherein pretext tasks derive supervision directly from input data, has become a cornerstone of visual representation learning. Early approaches included context-based tasks such as predicting relative patch positions (Doersch et al., 2015) and patch orderings (Noroozi & Favaro, 2016). While these works revealed the potential of such proxy tasks, they were limited in scalability. More recently, SSL has been dominated by two major families: (1) reconstruction-based methods (Zhou et al., 2021; He et al., 2022b; Bao et al., 2021; Assran et al., 2023) and (2) discriminative methods (He et al., 2020; Chen et al., 2020; Caron et al., 2021). These approaches have demonstrated impressive scalability and transferability, establishing strong foundations for large-scale vision pre-training (Oquab et al., 2023; Siméoni et al., 2025).

Parallel to these approaches, jigsaw pretext tasks explicitly formulate visual learning as an ordering problem, requiring the model to recover the spatial or temporal structure of visual inputs. Noroozi & Favaro (2016) pioneered the $3 \times 3$ image jigsaw puzzle, which was later extended for iterative refinements (Wei et al., 2019), domain generalization (Carlucci et al., 2019), and fine-grained reasoning (Du et al., 2020). Extensions to video include spatiotemporal jigsaws (Ahsan et al., 2019; Huo et al., 2021; Wang et al., 2022) that jointly exploit appearance and motion cues. Recent jigsaw-style variants (Caron et al., 2024; Wang et al., 2023; Zhai et al., 2022) have shown competitive results with contrastive learning and masked image modeling when properly designed and scaled.

The characteristics of jigsaw-style tasks also make them a good fit for understanding-based MLLMs, which are optimized for visual understanding with textual outputs rather than dense reconstruction. A visual jigsaw task thus provides a lightweight, verifiable objective that requires no additional generative modules. Building on these advantages, our work introduces Visual Jigsaw as a self-supervised post-training stage to enhance vision-centric perception in MLLMs across image, video, and 3D modalities.

### 2.2 MLLM VISUAL UNDERSTANDING

MLLMs (Hurst et al., 2024; Comanici et al., 2025; Bai et al., 2025; Zhu et al., 2025a) have rapidly advanced, achieving strong performance across diverse multimodal tasks. These improvements have largely stemmed from more powerful LLM backbones, better image resolution strategies, improved vision encoders, higher-quality training datasets, and post-training techniques. However, relatively little attention has been devoted to enhancing the *intrinsic visual understanding* of MLLMs. Most existing efforts rely on scaling data to indirectly improve perception-related tasks.

X-Former(Swetha et al., 2024) adds visual reconstruction objective in MLLM training but its target is to better extract and combine vision features from two vision encoders and only supervise the connector with the reconstruction objective. Recent works (Wang et al., 2025b;a) demonstrate that explicitly adding visual reconstruction objectives enhances visual understanding, but such approaches require introducing extra generative modules and objectives, have only been demonstrated in settings where the MLLM is trained jointly with reconstruction from the beginning, and have not been validated on stronger models like Qwen2.5-VL (Bai et al., 2025). Meanwhile, unified multi-modal models (UMMs) (Xie et al., 2025b; Chen et al., 2025c; Deng et al., 2025; Chen et al., 2025b) explore combining vision understanding and generation in one model, but it has only been shown that understanding benefits visual generation Xie et al. (2025a) while optimizing generative objectives sometimes harm understanding abilities Pan et al. (2025); Chen et al. (2025b). In contrast, we propose a lightweight, post-training self-supervised task that strengthens visual perception and understanding in MLLMs without altering the architecture.

### 2.3 MLLM RL POST-TRAINING

RL post-training has played a pivotal role in advancing LLMs. Early paradigms such as RLHF (Ouyang et al., 2022) and DPO (Rafailov et al., 2023) focused on improving alignment with human preferences, while recent developments like RLVR (Lambert et al., 2024; Shao et al., 2024) have been shown to substantially enhance reasoning capabilities. Inspired by this success, the MLLM community has begun to apply similar paradigms. Most works concentrate on strengthening multi-

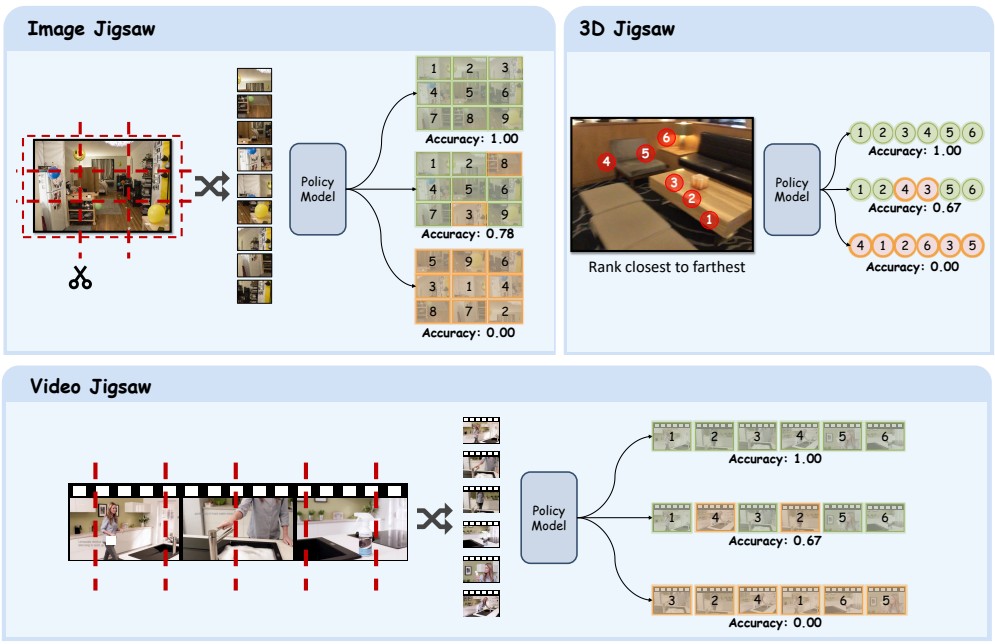

Figure 2: **Illustration of the Visual Jigsaw tasks.** In the Image Jigsaw (top left), an image is partitioned into non-overlapping patches, shuffled into a sequence, and the model is tasked with predicting the correct raster order. In the Video Jigsaw (bottom), a video is segmented into temporal clips, shuffled, and the model predicts their original chronological order. In the 3D Jigsaw (top right), points with distinct depth values are sampled from an RGB-D image, shuffled and annotated in the RGB view, and the model is required to recover the correct depth order from nearest to farthest. Across all tasks, the policy model outputs an ordering that is compared against the ground truth, and a partial accuracy reward is assigned when only some elements are correctly ordered.

modal reasoning for mathematical and scientific tasks (Meng et al., 2025; Huang et al., 2025; Yuan et al., 2025a; Wang et al., 2025f). These RL-based approaches have also been extended to video (Feng et al., 2025; Chen et al., 2025d) and 3D domains (Yuan et al., 2025b). Other methods focus on specific vision tasks such as grounding (Liu et al., 2025b) and segmentation (Liu et al., 2025a). More recent efforts (Zheng et al., 2025; Su et al., 2025) also explore teaching MLLMs to use vision tools, enabling models to think with images and better retrieve and perceive the visual input via operations like crop and zoom-in.

However, the majority of these approaches still target text-based reasoning, task-specific objectives, or visual tool-calling, rather than directly improving intrinsic visual perception. Vicrit (Wang et al., 2025e) and LLaVA-Critic-R1 (Wang et al., 2025d) enhance perception and reasoning by detecting errors in captions or judging textual responses, but their training signals are ultimately tied to text–image alignment instead of intrinsic visual signal understanding. The most closely related work, Jigsaw-R1 Wang et al. (2025g), also attempts to introduce a jigsaw task for MLLM post-training. However, its approach struggles even on simple $2 \times 2$ image jigsaws, thus focusing mainly on predicting the relative position of a pair of patches. In contrast, our method leverages the standard, more complex visual jigsaw tasks to systematically enhance MLLM perception, and we demonstrate its effectiveness not only on images but also across video and 3D modalities.

## 3 METHOD

### 3.1 VISUAL JIGSAW

Our proposed Visual Jigsaw framework (illustrated in Fig 2) is formulated as a general visual ordering problem. Given some data from a certain visual modality (image, video, or 3D), we derive a set of $K$ jigsaw elements by applying a modality-specific partitioning rule, such as splitting an image into patches, segmenting a video into clips, or sampling points in a 3D scene. These elements

are then shuffled, and the model is tasked with predicting their original structural arrangement. Formally, the model predicts a permutation of size $K$ as a list of indices, which is then compared against the ground-truth permutation. We optimize this task using the GRPO algorithm. The following subsections detail our reward design and describe the specific instantiations of Visual Jigsaw across each of the three visual modalities.

### 3.1.1 REWARD DESIGN

The ground-truth of the visual jigsaw tasks is a list of indices that is directly verifiable. Instead of assigning only a binary accuracy reward, we design a graded reward function. An output that exactly matches the ground-truth permutation receives an accuracy reward of 1. For a valid but partially correct permutation, the reward is the fraction of correctly placed indices, scaled by a discount factor $\gamma \in (0, 1)$. This discount penalizes incomplete solutions, preventing the model from overestimating partial matches while still providing learning signals. To avoid reward hacking (*e.g.* predicting the same index for all positions), any output that is not a valid permutation of size $K$ is assigned a reward of 0. Formally, the accuracy reward function is given by

$$Reward(o, g) = \begin{cases} 1, & \text{if } o = g \\ \gamma \cdot \frac{1}{K} \sum_{i=1}^{K} \mathbf{1}[o_i = g_i], & \text{if ValidPermutation}(o) \wedge o \neq g, \\ 0, & \text{otherwise} \end{cases}$$

where $o$ denotes the model's predicted permutation, $g$ the ground-truth permutation, $K$ the number of jigsaw pieces, $\gamma$ the discount factor for partial correctness, and ValidPermutation$(o)$ an indicator of whether $o$ is a valid permutation of size $K$.

Besides the accuracy reward, we also require the model to put its thinking process within `<think></think>` and the final answer within `<answer></answer>`. A format reward of 0.2 will be assigned to outputs following the correct format, while outputs with an incorrect format will receive 0 values for both format and accuracy rewards.

### 3.1.2 IMAGE JIGSAW

Given an input image $I \in \mathbb{R}^{H \times W \times 3}$, we first partition it into a grid of $m \times n$ non-overlapping patches, each of size $\frac{H}{m} \times \frac{W}{n}$. This produces $K = m \times n$ patches arranged in raster order (row-major, top-left to bottom-right):

$$\mathcal{P} = [p_1, p_2, \ldots, p_K], \quad p_i \in \mathbb{R}^{\frac{H}{m} \times \frac{W}{n} \times 3}.$$

We then apply a random permutation $\pi : \{1, 2, \ldots, K\} \rightarrow \{1, 2, \ldots, K\}$ that maps an original position index $i$ to its shuffled position $\pi(i)$. The shuffled sequence of patches can therefore be written as

$$\mathcal{P}_\pi = [\, p_{\pi^{-1}(1)}, \, p_{\pi^{-1}(2)}, \, \ldots, \, p_{\pi^{-1}(K)} \,],$$

where the $j$-th element corresponds to the patch originally at position $\pi^{-1}(j)$. Given $\mathcal{P}_\pi$, the model's objective is to recover the original arrangement by predicting the correct permutation of the input patch indices $[\pi(1), \pi(2), \ldots, \pi(K)]$.

For training, we use 118K images from the COCO dataset Lin et al. (2014). We set $m = n = 3$, yielding 9 patches per image, and we filter out images with side lengths smaller than 84 pixels to avoid overly small patches. The prompt template for this task is provided in Appendix A.6.

### 3.1.3 VIDEO JIGSAW

Given a video $V \in \mathbb{R}^{T \times H \times W \times 3}$ with $T$ frames, we segment it uniformly along the temporal axis into $K$ non-overlapping clips, each containing $\frac{T}{K}$ consecutive frames:

$$\mathcal{V} = [v_1, v_2, \ldots, v_K], \quad v_i \in \mathbb{R}^{\frac{T}{K} \times H \times W \times 3}.$$

We then apply a random permutation $\pi : \{1, 2, \ldots, K\} \rightarrow \{1, 2, \ldots, K\}$, where $\pi(i)$ denotes the shuffled position of the $i$-th clip in the original chronological order. The shuffled sequence is written as

$$\mathcal{V}_\pi = [\, v_{\pi^{-1}(1)}, \ v_{\pi^{-1}(2)}, \ \ldots, \ v_{\pi^{-1}(K)} \,].$$

The model's objective is to restore the original chronological order by predicting the correct permutation $[\pi(1), \pi(2), \ldots, \pi(K)]$.

For training, we use 100K videos from the LLaVA-Video dataset (Zhang et al., 2024c). Each video is divided into 6 clips ($K = 6$). To prevent the model from exploiting simple frame-matching cues at clip boundaries, we trim 5% of the frames from both the beginning and end of each clip. The maximum number of frames for each clip is set to 12, and the maximum resolution for each frame is set to $128 \times 28 \times 28$ pixels. We remove videos with lengths smaller than 24 seconds. The prompt template for this task can be found in Appendix A.6.

### 3.1.4 3D JIGSAW

A canonical 3D jigsaw task would mirror its 2D image and video counterparts, involving the partitioning of 3D space into volumetric primitives (*e.g.* voxels, mesh fragments, or point cloud segments) and tasking the model with recovering the original spatial arrangement. Such formulations would fully leverage geometric information in native 3D representations. However, current general-purpose MLLMs typically process 3D-related tasks via 2D images or videos rather than directly operating on raw 3D data structures.

We therefore design a practical variant of the 3D jigsaw based on RGB-D images. Given an RGB-D image, we randomly select $K$ points with distinct depth values, forming a sequence sorted by depth from nearest to farthest:

$$\mathcal{P} = [p_1, p_2, \ldots, p_K], \quad d_{p_1} < d_{p_2} < \cdots < d_{p_K},$$

where $d_{p_i}$ is the depth of point $p_i$.

We then apply a random permutation $\pi : \{1, 2, \ldots, K\} \rightarrow \{1, 2, \ldots, K\}$ to obtain a shuffled sequence of the points

$$\mathcal{P}_\pi = [\, p_{\pi^{-1}(1)}, \ p_{\pi^{-1}(2)}, \ \ldots, \ p_{\pi^{-1}(K)} \,].$$

Each point is annotated with its index in $\mathcal{P}_\pi$ on the RGB image. The model is tasked with recovering the correct depth order by predicting the permutation $[\pi(1), \pi(2), \ldots, \pi(K)]$ that restores $\mathcal{P}$.

For this task, we use the RGB-D data from ScanNet (Dai et al., 2017) and generate 300K training samples in total. We construct training samples by randomly selecting 6-point combinations from depth maps, restricting the points to lie within a range of 0.1 m to 10 m. To ensure diversity, any two points in a combination must be separated by at least 40 pixels in the image and differ in depth by more than 0.2 m. The prompt template for this task can be found in Appendix A.6. We also experimented with alternative designs of 3D jigsaw tasks, which are provided in Appendix A.2.

## 4 EXPERIMENTS

### 4.1 IMPLEMENTATION DETAILS

We adopt Qwen2.5-VL-7B-Instruct as the base MLLM for all experiments. We use the GRPO algorithm and remove both the KL regularization and the entropy loss. The discount factor $\gamma$ for partially correct predictions is set to 0.2. The training is performed with a global batch size of 256 for image jigsaw and 128 for video & 3D jigsaw, and the learning rate is $1 \times 10^{-6}$. For each prompt, we sample 16 responses with a decoding temperature of 1.0. Both image and video jigsaw tasks are trained for 1000 steps, and the 3D jigsaw is trained for 800 steps.

### 4.2 MAIN RESULTS

This section presents quantitative results. Qualitative examples are provided in Appendix A.4.

### 4.2.1 IMAGE JIGSAW

We evaluate the model trained with image jigsaw across three categories of vision-centric benchmarks including 1) **Fine-grained perception & understanding**: MMVP (Tong et al., 2024), fine-grained perception subset of MMStar (Chen et al., 2024), MMBench (Liu et al., 2024a), HR-Bench (Wang et al., 2025c), V* (Wu & Xie, 2024), MME-RealWorld (lite) (Zhang et al., 2025b), LISA-Grounding (Lai et al., 2024), OVD-Eval (Yao et al., 2024); 2) **Monocular spatial understanding**: VSR (Liu et al., 2023), OminiSpatial (Jia et al., 2025), DA-2K (Yang et al., 2024); 3) **Compositional visual understanding**: Winoground (Thrush et al., 2022), SugerCrepe++ (Dumpala et al., 2024).

We include three baselines, which are all post-trained from Qwen2.5-VL-7B. ThinkLite-VL (Wang et al., 2025f) mainly focuses on improving multimodal reasoning. VL-Cogito (Yuan et al., 2025a) is trained on a broader set of tasks, including general image understanding and counting, in addition to mathematical and scientific reasoning. LLaVA-Critic-R1 (Wang et al., 2025d) is trained with the critic task and shows improvement in image perception and understanding. As these vision-centric benchmarks mainly focus on direct visual perception and understanding, we directly evaluate the model to give the short answer without the thinking/reasoning process for fair comparison. This protocol is further motivated by our finding that enabling chain-of-thought reasoning can actually degrade the performance of some reasoning models on some specific benchmarks (*e.g.* 35.78 $\rightarrow$ 31.44 on OVD-Eval for ThinkLite-VL).

Table 1: **Evaluation results on image benchmarks.** Image Jigsaw achieves consistent improvements across fine-grained perception, spatial understanding, and compositional understanding tasks.

| Model | Fine-grained Perception & Understanding | | | | | | | | Spatial Und (Mono) | | | Compositional Und | |
| | MMVP | MMStar (fine-grained) | MMBench | HR-Bench-8K | V* | MME-RealWorld | LISA-Grounding | OVD-Eval | VSR | OmniSpatial | DA-2K | Winoground | SugarCrepe++ |
| | test | fine | en_dev | test | test | lite | test | test | test | test | val | g-acc | test |
| ThinkLite-VL | 55.33 | 59.95 | 84.19 | 68.12 | 76.96 | 46.17 | 73.70 | 35.78 | 78.09 | 42.60 | 58.46 | 35.25 | 61.49 |
| VL-Cogito | 55.33 | 56.64 | 82.98 | 69.62 | 79.58 | 47.63 | 72.26 | 35.78 | 79.82 | 44.29 | 56.43 | 38.25 | 63.59 |
| LLaVA-Critic-R1 | 53.33 | 57.80 | 83.16 | 67.50 | 78.01 | 45.18 | 68.52 | 35.28 | 78.50 | 42.73 | 53.82 | 34.75 | 61.93 |
| Qwen2.5-VL-7B | 54.66 | 59.75 | 83.33 | 67.38 | 76.96 | 43.41 | 71.89 | 35.07 | 77.68 | 42.66 | 54.45 | 37.00 | 61.59 |
| Image Jigsaw (SFT) | 56.00 | 60.94 | 83.67 | 69.75 | 80.10 | 43.88 | 66.59 | 34.35 | 80.68 | 43.55 | 61.46 | 38.75 | 62.03 |
| **Image Jigsaw** | 60.66 | 65.81 | 84.45 | 71.13 | 80.63 | 45.96 | 74.54 | 36.49 | 80.36 | 44.49 | 60.35 | 39.00 | 63.02 |
| *(Gain)* | +6.00 | +6.06 | +1.12 | +3.75 | +3.66 | +2.55 | +2.65 | +1.42 | +2.68 | +1.83 | +5.90 | +2.00 | +1.43 |

Tab 1 shows that our method consistently improves the vision-centric capabilities on the three types of benchmarks. These results confirm that incorporating image jigsaw post-training significantly enhances MLLMs' perceptual grounding and fine-grained vision understanding beyond reasoning-centric post-training strategies. We attribute these improvements to the fact that solving image jigsaw requires the model to attend to local patch details, infer global spatial layouts, and reason about inter-patch relations, which directly benefits fine-grained, spatial, and compositional understanding.

### 4.2.2 VIDEO JIGSAW

For video jigsaw, we evaluate on a comprehensive suite of video benchmarks: AoTBench (Xue et al., 2025), Vinoground (Zhang et al., 2024a), TOMATO (Shangguan et al., 2024), FAVOR-Bench (Tu et al., 2025), TUNA-Bench (Kong et al., 2025), Video-MME (Fu et al., 2025), TempCompass (Liu et al., 2024b), TVBench (Cores et al., 2024), MotionBench (Hong et al., 2025), LVBench (Wang et al., 2024b), VSI-Bench (Yang et al., 2025), Video-TT (Zhang et al., 2025c), CVBench (Zhu et al., 2025b).

We include the Video-R1 (Feng et al., 2025) baseline for comparison, which is trained with cold-start SFT followed by RL for video understanding and reasoning. We enable the thinking process when evaluating Video-R1, as we find its performance is generally better than direct answering. For

all models, we set the maximum number of pixels to $256 \times 28 \times 28$ and evaluate under three different frame settings (16, 32, 64).

Table 2: **Evaluation results on video benchmarks.** Video Jigsaw consistently improves over the baseline across all benchmarks and frame settings.

| Model | Frames | AoTBench | Vinoground | TOMATO | FAVOR-Bench | TUNA-Bench | VideoMME | TempCompass | TVBench | MotionBench | LVBench | VSI-Bench | Video-TT | CVBench |
|---|---|---|---|---|---|---|---|---|---|---|---|---|---|---|
| | | vqa | group | test | test | test | wo subs | mc | test | val | test | test | mcq | test |
| Video-R1 | 16 | 45.06 | 9.40 | 27.29 | 49.47 | 53.00 | 56.62 | 70.19 | 51.80 | 55.82 | 34.53 | 34.34 | 42.95 | 47.50 |
| Video-R1 | 32 | 47.53 | 10.20 | 27.29 | 49.90 | 54.26 | 59.88 | 71.77 | 53.54 | 56.12 | 38.61 | 35.11 | 42.63 | 48.10 |
| Video-R1 | 64 | 48.68 | 10.60 | 27.36 | 50.51 | 54.33 | 60.85 | 72.59 | 53.43 | 56.09 | 38.80 | 36.61 | 42.74 | 48.69 |
| Qwen2.5-VL-7B | 16 | 45.52 | 12.60 | 25.87 | 48.54 | 53.14 | 57.44 | 71.77 | 49.94 | 55.56 | 33.51 | 32.79 | 38.39 | 47.70 |
| Qwen2.5-VL-7B | 32 | 49.48 | 18.20 | 26.34 | 49.34 | 54.88 | 60.70 | 72.59 | 51.96 | 56.47 | 39.19 | 35.34 | 41.57 | 49.60 |
| Qwen2.5-VL-7B | 64 | 52.41 | 21.80 | 26.35 | 50.86 | 55.79 | 63.44 | 72.84 | 53.74 | 56.29 | 40.35 | 37.74 | 42.25 | 51.50 |
| **Video Jigsaw** | 16 | 51.67 | 15.20 | 27.56 | 49.69 | 55.10 | 58.07 | 73.10 | 51.33 | 56.87 | 36.41 | 35.39 | 40.19 | 49.80 |
| *(Gain)* | | +6.15 | +2.60 | +1.69 | +1.15 | +1.96 | +0.63 | +1.33 | +1.39 | +1.31 | +2.90 | +2.60 | +1.80 | +2.10 |
| **Video Jigsaw** | 32 | 55.00 | 21.40 | 28.03 | 50.56 | 56.49 | 62.37 | 73.60 | 53.31 | 57.99 | 39.70 | 38.47 | 43.27 | 51.60 |
| *(Gain)* | | +5.52 | +3.20 | +1.69 | +1.22 | +1.61 | +1.67 | +1.01 | +1.35 | +1.52 | +0.51 | +3.13 | +1.70 | +2.00 |
| **Video Jigsaw** | 64 | 57.64 | 25.20 | 28.30 | 52.27 | 56.63 | 64.74 | 73.60 | 54.18 | 57.91 | 41.83 | 40.40 | 44.11 | 54.50 |
| *(Gain)* | | +5.23 | +3.40 | +1.95 | +1.41 | +0.84 | +1.30 | +0.76 | +0.44 | +1.62 | +1.48 | +2.66 | +1.86 | +3.00 |

From the results shown in Tab 2, we observe that Video Jigsaw brings consistent improvements across all video understanding benchmarks and frame settings. While our method enhances general video perception and comprehension, the gains are particularly pronounced on tasks requiring temporal-centric understanding and reasoning about temporal directionality (*e.g.* AoTBench). Furthermore, the strong gains on CVBench demonstrate improved cross-video understanding and reasoning. These results confirm that solving video jigsaw tasks encourages the model to better capture temporal continuity, understand relationships across videos, reason about directional consistency, and generalize to holistic and generalizable video understanding scenarios.

### 4.2.3 3D JIGSAW

For the 3D modality, we evaluate the model on a diverse set of benchmarks that span various aspects of 3D understanding: SAT-Real (Ray et al., 2024), 3DSRBench (Ma et al., 2024), ViewSpatial (Li et al., 2025), All-Angles (Yeh et al., 2025), OminiSpatial (Jia et al., 2025), VSI-Bench (Yang et al., 2025), SPARBench (tiny) (Zhang et al., 2025a), and DA-2K (Yang et al., 2024).

Table 3: **Evaluation results on 3D benchmarks.** 3D Jigsaw consistently enhances performance on both directly related depth comparison tasks (DA-2K) and broader 3D perception tasks spanning single-view, multi-view, and egocentric video inputs.

| Model | SAT-Real | 3DSRBench | ViewSpatial | All-Angles | OmniSpatial | VSI-Bench | SPARBench | DA-2K |
|---|---|---|---|---|---|---|---|---|
| | test | test | test | test | test | test | tiny | test |
| Qwen2.5-VL-7B | 48.66 | 57.42 | 36.52 | 47.56 | 42.66 | 37.74 | 35.75 | 54.45 |
| **3D Jigsaw** | 64.00 | 58.13 | 38.62 | 49.06 | 45.99 | 40.64 | 38.31 | 71.56 |
| *(Gain)* | +15.34 | +0.71 | +2.10 | +1.50 | +3.33 | +2.90 | +2.56 | +17.11 |

As shown in Tab 3, 3D Jigsaw achieves significant improvements across all benchmarks. Unsurprisingly, the largest gain is on DA-2K, a depth estimation benchmark that is directly related to our depth-ordering pre-training task. More importantly, we observe consistent improvements on a wide

range of other tasks, including those with single-view (*e.g.* 3DSRBench, OminiSpatial), multi-view (*e.g.* ViewSpatial, All-Angles), and egocentric video inputs (*e.g.* VSI-Bench). These results demonstrate that our approach not only teaches the specific skill of depth ordering but also effectively strengthens the model's general ability to perceive and reason about 3D spatial structure.

## 4.3 ABLATION STUDIES AND DISCUSSIONS

**SFT vs. RL.** We investigate the difference between using SFT and RL to train the visual jigsaw task, focusing on the image jigsaw setting. As shown in the Image Jigsaw (SFT) entry of Tab 1, SFT leads to moderate improvements on some benchmarks, but the gains are notably smaller than those achieved with RL. Moreover, on certain benchmarks (*e.g.* LISA-Grounding and OVD-Eval), SFT causes a significant performance degradation, suggesting that the model overfits to the jigsaw task and fails to transfer the learned skills. This observation is consistent with recent findings that SFT tends to memorization, while RL is better at promoting generalization (Huan et al., 2025; Chu et al., 2025). Our results confirm that RL enables the model to more effectively generalize the vision-centric capabilities acquired from visual jigsaw training to related downstream tasks.

**How does the difficulty of the visual jigsaw tasks affect the performance?**

We conducted ablation experiments to investigate how the difficulty of the jigsaw task affects model performance. We varied the complexity of both the image and video jigsaw tasks: for the image task, we reduced the grid size from $3\times3$ to $2\times2$; for the video task, we reduced the number of clips from six to four. We then measured the average performance across all corresponding benchmarks (using 16 frames for the video evaluation). The results in Fig 3 show that while easier jigsaw tasks still yield perfor-

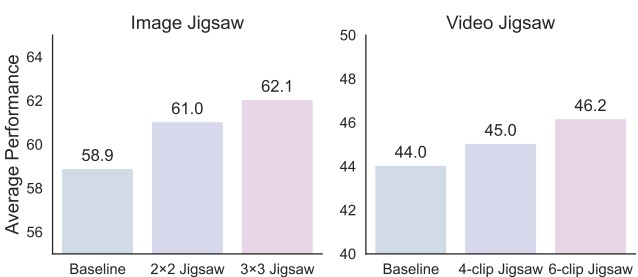

Figure 3: Performance with different jigsaw difficulties on image and video tasks.

mance improvements over the baseline, the gains are substantially smaller than those from the standard, more difficult tasks. This indicates that a higher degree of difficulty provides a stronger supervisory signal for enhancing fine-grained perception and temporal reasoning. Critically, we also found that on challenging setups (*e.g.* $3 \times 3$ for images), the design of the partial accuracy reward becomes crucial. Without this design, the model fails to learn the task, as sparse binary feedback is insufficient to bootstrap learning in the early stages of training.

**Visual Jigsaw on other base models.** To validate the general effectiveness of our Visual Jigsaw framework, we apply it on a stronger base model MiMo-VL-7B-SFT-2508 (Xiaomi, 2025) and show the results in Table 4, where Visual Jigsaw also yields consistent gains over the baseline.

| Model | Image Avg | Video Avg | 3D Avg |
|---|---|---|---|
| MiMo-VL-7B-SFT | 63.77 | 51.84 | 50.67 |
| + Visual Jigsaw | 65.14 | 54.47 | 52.91 |

Table 4: Results of Visual Jigsaw on MiMo-VL.

**Visual Jigsaw on Reasoning MLLMs.**

**Harder Jigsaw Configurations.** We also conduct experiments on harder jigsaw configurations with $4 \times 4$ for image jigsaw and 8-clip for video jigsaw. For $4 \times 4$ image jigsaw, We find that the model failed to learn this task effectively, with jigsaw accuracy remaining low. This happens even when we do cross-configuration adaptation by first training on $3 \times 3$ jigsaw task. We attribute this to three key factors: (1) Information Scarcity: At the typical resolution of our COCO training data ($\approx$640$\times$480), $4 \times 4$ tiles are often too small and fragmented to contain sufficient semantic information for re-assembly. (2) Semantic Ambiguity: Small tiles from uniform areas (e.g., sky, walls) become perceptually indistinguishable, making the ordering task ill-posed. (3) Combinatorial Explosion. The permutation space expands from 9! to 16!, a significant increase in complexity for the model. This suggests that scaling to harder configurations is not merely a matter of increasing grid size, but will likely require a combination of higher-resolution training data, tile-diversity con-

straints, and curriculum learning strategies. As for the 8-clip video jigsaw, the average performance is 46.24 with 16-frame setting which does not bring notable improvement over 6-clip. This is mainly because many LLaVA-Video samples are around 30 s, so splitting into 8 clips makes segments short and sometimes ambiguous. We leave the study 8-clip on longer videos for future works.

Table 5: Performance of Visual Jigsaw on a reasoning-oriented MLLM (ThinkLite-VL), showing improved visual perception while preserving reasoning ability.

| Model | Vision-Centric Avg | MathVista testmini | MathVision testmini | MathVerse testmini | MMMU val | EMMA mini |
|---|---|---|---|---|---|---|
| ThinkLite-VL | 59.69 | 75.20 | 30.92 | 50.76 | 55.11 | 26.75 |
| + Image Jigsaw | 61.60 | 75.10 | 35.20 | 50.50 | 54.22 | 29.00 |

We further explore whether Visual Jigsaw can also benefit reasoning MLLMs that have already undergone reasoning-intensive RL post-training. To this end, we select ThinkLite-VL as the base model and apply the image jigsaw training. We enable the KL constraint to better preserve the reasoning capability during training. We evaluate the resulting model on both vision-centric benchmarks and multimodal reasoning benchmarks, including MathVista (Lu et al., 2023), MathVision (Wang et al., 2024a), MathVerse (Zhang et al., 2024b), MMMU (Yue et al., 2024), and EMMA(Hao et al., 2025). As shown in Tab 5, the reasoning MLLM trained with Visual Jigsaw achieves clear improvements in visual perception and understanding, while maintaining its strong reasoning ability.

**How does model solve jigsaw problems?** Here we provide an example think trace for image jigsaw training in Fig 4. We can see that the model identifies semantic objects (trees, sky, a person) in different image tiles, infers spatial context (horizon line, pathway edges), and uses logical deduction to place the tiles into a coherent global scene. More examples are provided in Appendix A.1.

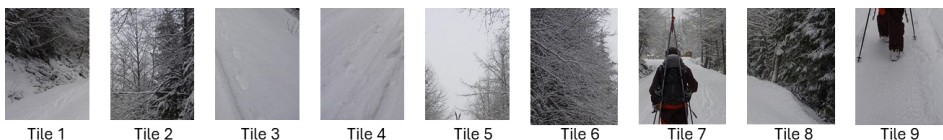

Tile 1   Tile 2   Tile 3   Tile 4   Tile 5   Tile 6   Tile 7   Tile 8   Tile 9

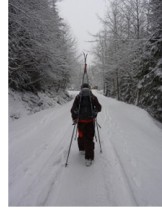

<think>
I am mentally reassembling the image based on the spatial relationships and features of the tiles. Tile 2 is positioned at the upper right corner, showing a dense cluster of snow-covered trees with a dark, shadowed area to its right, suggesting the edge of a forested area or a hillside. Tile 5, located at the top-left portion of the image, appears to be part of a snowy, overcast sky with tree branches extending from the left side, indicating a continuation of the snowy landscape with a clear horizon line. Tile 1, situated at the bottom-left and lower middle-left areas, shows a snowy slope with a darker, shaded section that transitions into a lighter, snow-covered ground, consistent with a path or trail leading into a wooded area. Tile 7, occupying the central and right-middle parts of the image, depicts a person wearing winter gear, standing on a snowy path with ski poles, likely engaged in skiing or walking through a snowy environment. Tiles 1 and 8 appear to form the sides of this snowy pathway, with tile 1 on the left and tile 8 on the right, framing the central figure and the trail. This arrangement suggests a continuous snowy outdoor scene with a focus on a person navigating a snowy terrain.
</think>
<answer> 6, 5, 2, 1, 7, 8, 4, 9, 3</answer>

Figure 4: Example thinking process for solving image jigsaw.

## 5  CONCLUSION

In this work, we introduced Visual Jigsaw, a verifiable self-supervised post-training framework that enhances vision-centric understanding in MLLMs. By formulating visual understanding as an ordering problem and optimizing it with RLVR, visual jigsaw avoids the need for dense visual reconstruction and integrates seamlessly into text-only MLLMs. Our experiments demonstrate the generality of this approach, yielding consistent improvements across images, videos, and 3D data in fine-grained perception, temporal reasoning, and 3D spatial understanding. Ultimately, our work highlights the potential of perception-focused self- and weakly-supervised tasks as a powerful and complementary path toward developing more capable and robust multimodal models.

ETHICS STATEMENT

This work uses only publicly available datasets that follow established licenses and guidelines. Our method focuses on improving vision-centric perception and understanding in MLLMs without introducing additional risks beyond existing models. As with other MLLMs, potential misuse or biases may arise if training data are not carefully curated. We emphasize responsible usage, transparency, and alignment with human intentions to maximize benefits while mitigating risks.

REPRODUCIBILITY STATEMENT

All datasets used in the experiments are publicly available, and we provide detailed descriptions of the used datasets and preprocessing information in Sec 3. The evaluation benchmarks and details are provided for each modality in Sec 4. The training setup and hyperparameters are described in Sec 4.1. Our implementation is based on the open-source `verl` (Sheng et al., 2024) library, with the main modifications including the construction of visual jigsaw data and the reward calculation. The corresponding code is provided in the supplementary materials. To further facilitate reproducibility, we will release the code, data, and models to reproduce all main experiments, along with instructions for running ablation studies and evaluations.

ACKNOWLEDGEMENT

This research is supported by cash and in-kind funding from NTU S-Lab and industry partner(s). This study is also supported by the Ministry of Education, Singapore, under its MOE AcRF Tier 2 (MOE-T2EP20221-0012, MOE-T2EP20223-0002).

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

## A APPENDIX

### USE OF LARGE LANGUAGE MODELS

In this work, we have used large language models to assist in polishing the writing of the paper. Specifically, the LLM was employed to check grammar correctness and to provide alternative phrasings or stylistic suggestions for certain sentences. All suggestions were carefully reviewed by the authors, and only adopted after manual verification and modification when appropriate.

### A.1 QUALITATIVE EXAMPLES OF THINKING PROCESS IN VISUAL JIGSAW

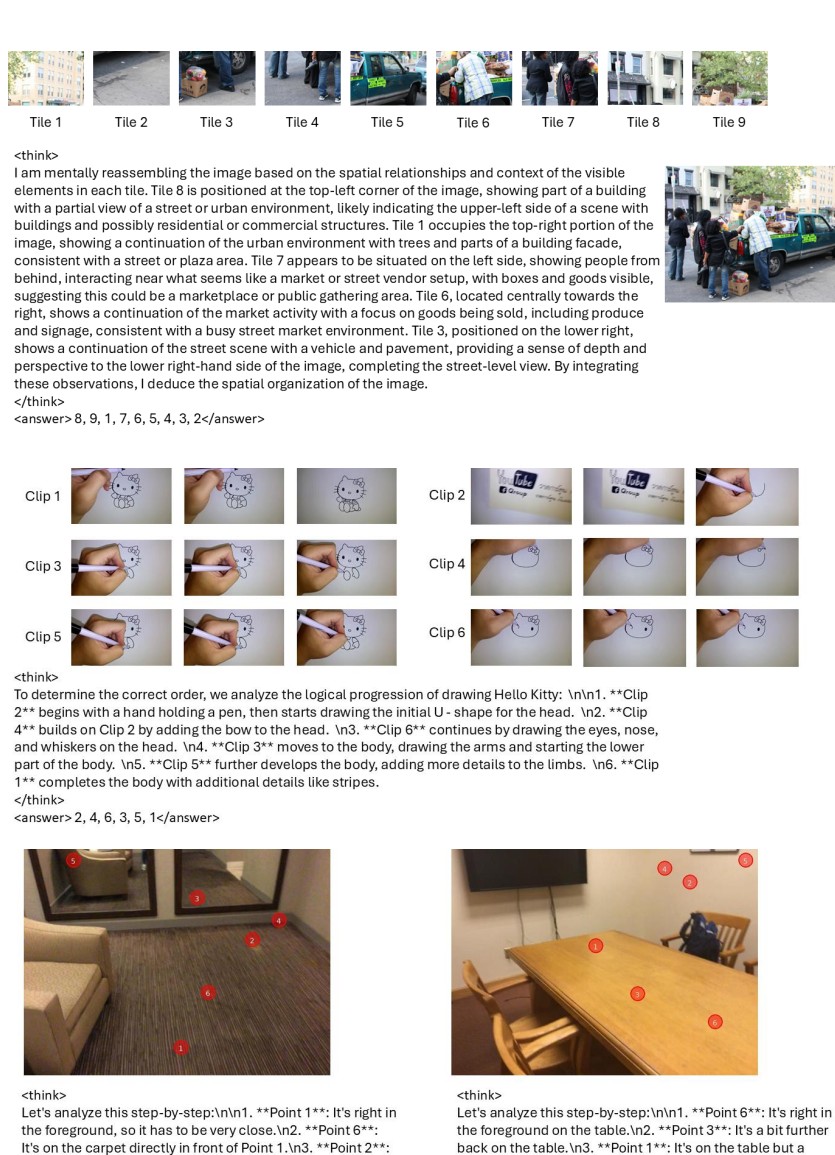

Figure 5: Examples of the thinking process solving visual jigsaw tasks.

In this section, we provide an example thinking process of our model solving the visual jigsaw task in Fig 5. For the image jigsaw task, the model needs to identify the content for each image patch,

imagine the overall scene, and reason over the relationships between the image patches to solve the jigsaw puzzle. For the video jigsaw, the model identifies the content for each clip and reasons about the correct chronological order. As for the 3D jigsaw, the model grounds the point marker on the actual object, infers the distance between points and the camera, and compares the distances among the points.

## A.2  ADDITIONAL EXPERIMENTS ON 3D JIGSAW

Besides the depth ordering task for the 3D modality, we have also explored two variants of 3D jigsaw designs, which we detail below.

**(1) View–Motion Matching.** Given a scene captured from multiple camera poses, we randomly select one view as the anchor view, and sample several candidate views that differ from the anchor while also being diverse from one another. For each candidate, we construct a natural language description of the ego-motion from the anchor to that candidate (*e.g.* "move forward 2.0 meters and rotate left by $15°$"). The model is provided with the anchor image, the shuffled candidate images, and the corresponding ego-motion descriptions, and is tasked with correctly matching each candidate view to its ego-motion description.

**(2) BEV–Pose Matching.** We render a bird's-eye-view (BEV) image of the scene and randomly select a set of candidate views with different camera poses. These camera poses are annotated on the BEV image with numerical identifiers. The model is then given the annotated BEV image and a shuffled set of candidate view images, and must correctly match each camera pose to its corresponding candidate view.

Table 6: **Evaluation of 3D Jigsaw Variants.** Comparison of depth ordering, view–motion matching, and BEV–pose matching tasks on 3D benchmarks.

| Model | SAT-Real | 3DSRBench | ViewSpatial | All-Angles | OmniSpatial | VSI-Bench | SPARBench | DA-2K |
|---|---|---|---|---|---|---|---|---|
| | test | test | test | test | test | test | tiny | test |
| Qwen2.5-VL-7B | 48.66 | 57.42 | 36.52 | 47.56 | 42.66 | 37.74 | 35.75 | 54.45 |
| Depth Ordering | 64.00 | 58.13 | 38.62 | 49.06 | 45.99 | 40.64 | 38.31 | 71.56 |
| View–Motion Matching | 64.67 | 55.89 | 37.94 | 48.22 | 44.55 | 38.97 | 36.53 | 60.15 |
| BEV–Pose Matching | 62.00 | 57.17 | 36.69 | 48.22 | 44.22 | 38.78 | 34.31 | 58.99 |

**Analysis and Results.** Both tasks are intuitively reasonable, as they explicitly encourage the model to connect 2D visual observations with underlying 3D spatial configurations. However, our preliminary experiments (shown in Tab 6) show that these variants do not lead to significant improvements on downstream benchmarks, and overall underperform the depth ordering formulation. We hypothesize that this may be due to the relatively weak 3D perception and reasoning capability of current base MLLMs, which limits their ability to transfer the learned skills from these complex jigsaw formulations to downstream tasks. Exploring ways to strengthen this foundation and better exploit such 3D-aware self-supervised tasks remains an interesting direction for future work.

## A.3  VISUAL JIGSAW EXAMPLES

The visual jigsaw task examples for the three modalities are provided in Fig. 6, Fig. 7, and Fig. 8.

## A.4  QUALITATIVE EXAMPLES

Some qualitative examples of the model trained with image, video, and 3D jigsaw tasks are shown in Fig. 9, Fig. 10, and Fig. 11.

**Image Jigsaw**
**Mentally reconstruct the image and output the patch indices in the correct raster scan order.**

Figure 6: Examples of the image jigsaw task. Each row shows a shuffled set of patches from an image, where the model is required to reconstruct the correct raster scan order. The ground-truth answers are displayed on the right.

## A.5 LIMITATIONS AND FUTURE WORKS

While our study demonstrates the effectiveness of Visual Jigsaw across images, videos, and 3D modalities, several limitations remain. First, for both image and video jigsaw, we adopt a standard and relatively simple formulation. Future work could explore more complex or hybrid jigsaw configurations, such as jointly partitioning video along spatial and temporal dimensions, or using heterogeneous piece sizes to introduce richer structural constraints. Second, due to computational constraints, we have not scaled the training data and model size extensively; investigating the scalability of this self-supervised task remains a promising direction. Third, some of our 3D jigsaw variants did not yield the expected improvements. We believe that applying these tasks to base models with stronger 3D reasoning capabilities and richer 3D priors could unlock further potential. Finally, beyond jigsaw, it is worth exploring a broader range of self- and weakly-supervised vision-centric tasks to enhance the perceptual and reasoning abilities of multimodal large language models.

**Video Jigsaw**
**Mentally reconstruct the video and output the clip indices in the correct chronological order.**

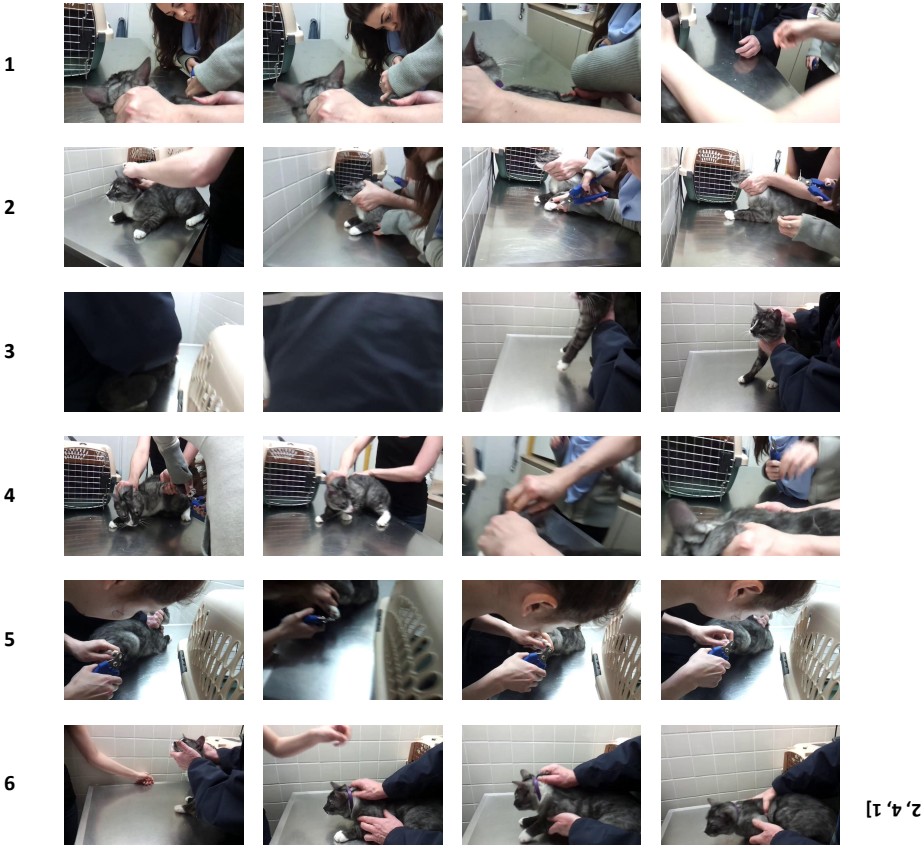

Figure 7: Example of the video jigsaw task. Each row shows a clip from the original video image, and the 6 clips are shuffled. The model is required to reconstruct the correct chronological order. The ground-truth answers are displayed on the right.

**3D Jigsaw**
**Order the points from closest to farthest relative to the camera.**

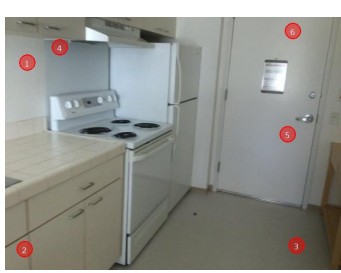

Answer
[2, 1, 4, 3, 6, 5]

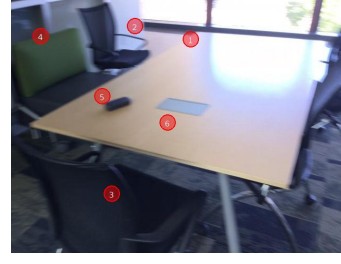

Answer
[3, 6, 5, 4, 1, 2]

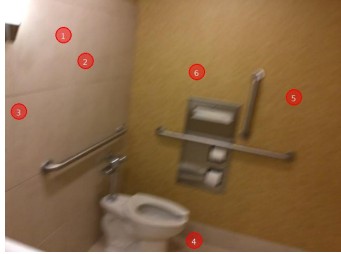

Answer
[3, 1, 2, 5, 6, 4]

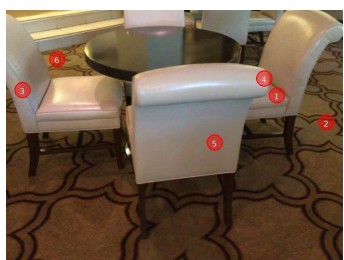

Answer
[4, 5, 3, 1, 2, 6]

Figure 8: Examples of the 3D jigsaw task. The model is required to order the points in each image from closest to farthest relative to the camera. The ground-truth answers are displayed on the right.

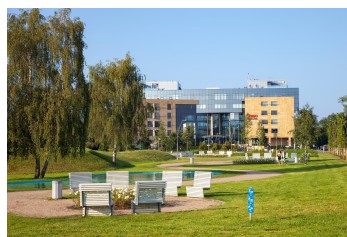

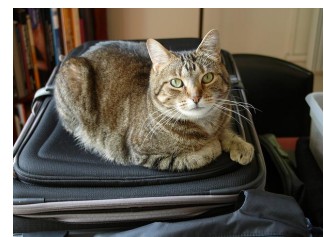

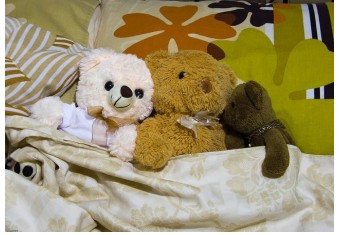

**Fine-grained Perception**
*What is the color of the woman's handbag?*
Qwen2.5-VL-7B: Black.
Image Jigsaw: White.

**Spatial Understanding**
*Decide whether the caption correctly describes the image. Caption: The suitcase is under the cat.*
Qwen2.5-VL-7B: No.
Image Jigsaw : Yes.

**Compositional Understanding**
*Does the caption describe the image correctly? Caption: A white, beige and brown baby bear under a beige/white comforter.*
Qwen2.5-VL-7B: No.
Image Jigsaw : Yes.

Figure 9: Qualitative examples on image tasks.

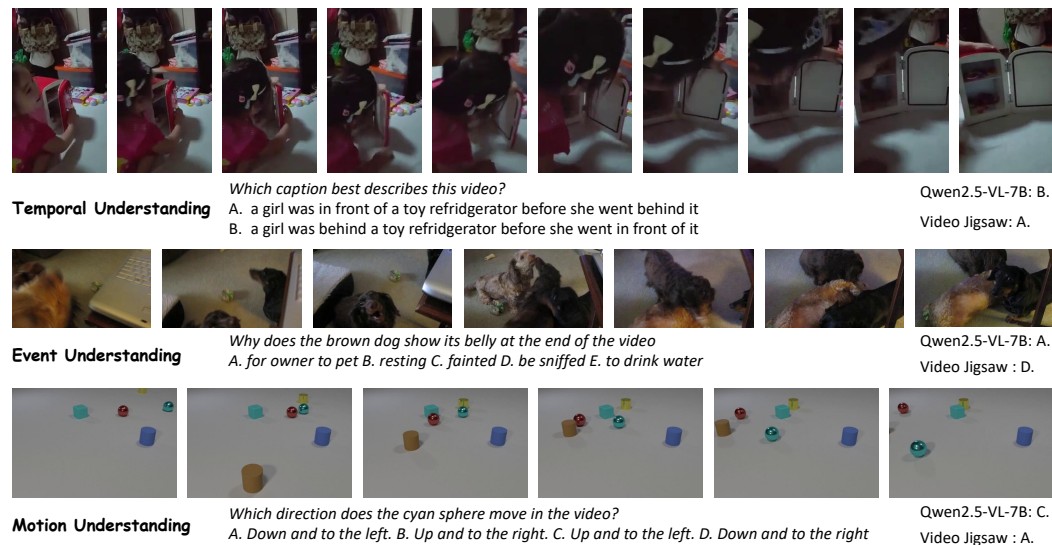

**Temporal Understanding**

*Which caption best describes this video?*
A. a girl was in front of a toy refridgerator before she went behind it
B. a girl was behind a toy refridgerator before she went in front of it

Qwen2.5-VL-7B: B.
Video Jigsaw: A.

**Event Understanding**

*Why does the brown dog show its belly at the end of the video*
A. for owner to pet B. resting C. fainted D. be sniffed E. to drink water

Qwen2.5-VL-7B: A.
Video Jigsaw : D.

**Motion Understanding**

*Which direction does the cyan sphere move in the video?*
A. Down and to the left. B. Up and to the right. C. Up and to the left. D. Down and to the right

Qwen2.5-VL-7B: C.
Video Jigsaw : A.

Figure 10: Qualitative examples on video tasks.

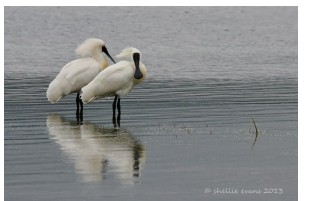

**Perspective Taking**

*How many cranes have their heads facing their own right?*
*A. One B. two C. three D. four*

Qwen2.5-VL-7B: B

3D Jigsaw: A

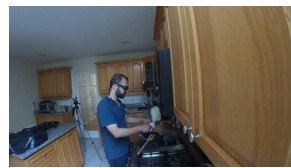

**Distance Comparison**

*View 1 and View 2 are two different views that represent the same scene. In which view, the milk carton in the image is closer to the spot where the camera view was positioned?*
*A. closer to the spot where camera View 1 was positioned B. closer to the spot where camera View 2 was positioned C. distance to the spot where camera View 1 and View 2 were positioned is equal D.*

Qwen2.5-VL-7B: A          3D Jigsaw : B

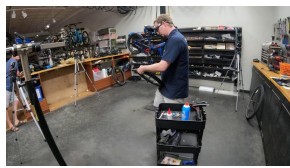
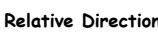

**Relative Direction**

*View 1 and View 2 are two different views that represent the same scene. In View 1, the man holding the bicycle tire is facing the left side of the view, which direction is the man holding the bicycle tire facing in View 2?*
*A. facing away from the spot where camera View 2 was positioned B. facing to the left side of View 2 C. facing to the spot where camera View 2 was positioned*

Qwen2.5-VL-7B: C          3D Jigsaw: A

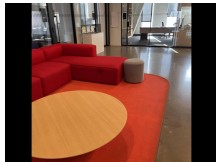
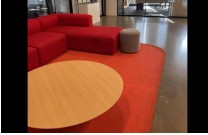
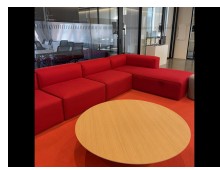

**Camera Movement**

*How did the camera likely rotate when shooting the video?*
*A. rotated left B. rotated right*

Qwen2.5-VL-7B: B

3D Jigsaw: A

Figure 11: Qualitative examples on 3D tasks.

A.6 TASK PROMPTS

---

**Prompt for Image Jigsaw**

You are given nine shuffled image tiles that were created by slicing one image into a 3*3 grid.

Here are the tiles, each tagged with an index reflecting the current (shuffled) order in which they are shown:

Tile 1: <image>
Tile 2: <image>
Tile 3: <image>
Tile 4: <image>
Tile 5: <image>
Tile 6: <image>
Tile 7: <image>
Tile 8: <image>
Tile 9: <image>

Task:
Mentally reassemble the original image, arranging the tiles into the correct 3*3 layout and provide the tile indices in raster-scan order (left-to-right, top-to-bottom), separated by commas.

Answer format example:
5, 1, 3, 7, 9, 2, 4, 8, 6
You FIRST think about the reasoning process as an internal monologue and then provide the final answer. The reasoning process MUST BE enclosed within <think></think> tags. The final answer MUST BE put within <answer></answer> tags.

---

**Prompt for Video Jigsaw**

You are given four **shuffled** video clips that were created by slicing one original video into 6 equal-length temporal segments.

Here are the clips, each tagged with an index reflecting the current (shuffled) order in which they are shown:

Clip 1: <video>
Clip 2: <video>
Clip 3: <video>
Clip 4: <video>
Clip 5: <video>
Clip 6: <video>

Task:
1. Mentally reassemble the original video by arranging the clips in their correct chronological order (earliest segment first, latest segment last).
2. Output the clip indices in that order, separated by commas.

Answer format example:
2, 3, 1, 4, 6, 5
You FIRST think about the reasoning process as an internal monologue and then provide the final answer. The reasoning process MUST BE enclosed within <think></think> tags. The final answer MUST BE put within <answer></answer> tags.

---

**Prompt for 3D Jigsaw**

<image>
You are given an indoor RGB image. Six points are marked on the image with red circular labels (1, 2, 3, . . . ).

Your task is to order the points from closest to farthest relative to the camera, judging the distance based on the center of the red circular marker.

Answer with the ordered sequence of point numbers.
You FIRST think about the reasoning process as an internal monologue and then provide the final answer. The reasoning process MUST BE enclosed within <think></think> tags. The final answer MUST BE put within <answer></answer> tags.

---

