# OpenReview forum: "Visual Jigsaw Post-Training Improves MLLMs"
_ICLR.cc/2026/Conference — ICLR 2026 Poster_

### Official Review · Reviewer_kYxc · 2025-10-15

**Soundness:** 3
**Presentation:** 2
**Contribution:** 2
**Rating:** 6
**Confidence:** 5

**Summary:**

This paper considers Visual Jigsaw as a general self-supervised task for post-training. Specifically, it has three variants, including image jigsaw, 3D jigsaw (ranking from the closest to the farthest), and video jigsaw. Empirically, initialized from Qwen2.5-VL-7B-Instruct, the proposed method achieves improvements over baselines separately. Moreover, when incorporating with a text-centric reasoning MLLM (ThinkLite-VL), the proposed image jigsaw post-training task still brings improvements on vision-centric benchmarks and maintains the performance on reasoning-oriented benchmarks.

**Strengths:**

1. This paper is well-written and easy to follow.
2. The motivation of this paper is clear and reasonable.
3. Empirical studies are partially sufficient.

**Weaknesses:**

1. The proposed three types of jigsaw tasks are evaluated *separately*. It is strongly encouraged to combine them and find out whether they can work together.
2. Qualitative examples of the reasoning pathways of both (1) when solving the proposed jigsaw problems and (2) answering multiple-choice questions are missing. Moreover, the underlying reason why this pre-text task is beneficial for VQA seems to be missing. An analysis of the reasoning pathways might be helpful.

Some suggestions on the claims:
1. The claim "While these jigsaw-style approaches provide structural ordering signals, they have generally shown weaker performance compared to more dominant approaches" is actually wrong. There are actually some great works [1, 2, 3] that demonstrate jigsaw-like self-supervised pre-training tasks are at least comparable with contrastive learning and masked image modeling. Therefore, adding discussions on these works and clarifying the claim is important.
2. The claim"Current post-training paradigms are predominantly text-centric, where dense visual inputs are only leveraged to extract sparse cues for text-based reasoning" is also not that appropriate. Recent o3-like approaches are definitely not text-centric reasoning methods. Discussions should include both early approaches like CogCoM [4] and Dyfo [5], recent approaches like DeepEyes [6], VGR [7], Pixel-Reasoner [8], and TreeVGR [9]. The advantages of visual jigsaw compared with these o3-like approaches should be discussed.

**References**

[1] Position prediction as an effective pretraining strategy. ICML, 2022

[2] DropPos: Pre-Training Vision Transformers by Reconstructing Dropped Positions. NeurIPS, 2023.

[3] Location-aware self-supervised transformers. WACV, 2024.

[4] CogCoM: A Visual Language Model with Chain-of-Manipulations Reasoning. ICLR 2025.

[5] Dyfo: A training-free dynamic focus visual search for enhancing lmms in fine-grained visual understanding. CVPR 2025.

[6] DeepEyes: Incentivizing" Thinking with Images" via Reinforcement Learning. arXiv 2025.

[7] Vgr: Visual grounded reasoning. arXiv 2025.

[8] Pixel reasoner: Incentivizing pixel-space reasoning with curiosity-driven reinforcement learning. arXiv 2025.

[9] Traceable evidence enhanced visual grounded reasoning: Evaluation and methodology. arXiv 2025.

**Questions:**

N/A

---

> ### Author Response · Authors · 2025-11-18
> **Response to reviewer kYxc (Part 1)**
>
> We thank the reviewer for the valuable feedback and concrete suggestions for improvement. Below, we address each point raised.
>
> ---
>
> > **Q1: Combine three types of jigsaw tasks.**
>
> **A1:** Great suggestion! Our initial separate evaluations were designed to carefully isolate the impact of our task and optimize its design for each modality.
> To address your question directly, we have now conducted a new multi-task learning experiment, jointly post-training a model on a mixture of image and 3D jigsaw tasks. (We excluded the more computationally intensive video jigsaw task due to time constraints but see this as a clear next step). The results are presented below:
>
> | Model | Image Benchmark Avg | 3D Benchmark Avg |
> | :--- | :---: | :---: |
> | Qwen2.5-VL-7B (Baseline) | 58.91 | 45.09 |
> | + Image Jigsaw (Specialist) | 62.06 | - |
> | + 3D Jigsaw (Specialist) | - | 50.78 |
> | **+ Combined (Image & 3D)** |62.28 | 50.56 |
>
> The multi-task model retains the similar gains achieved by the specialist models, showing that the underlying skills learned are compatible and can be developed in a unified training process.
>
> This experiment confirms that our Visual Jigsaw framework is robust to multi-task settings. We will add these results to the paper and agree that further exploring the combination of all three modalities is a promising direction for future work.
>
> > **Q2: Claims on jigsaw-like self-supervised pre-training tasks.**
>
> **A2:** Thank you for this important correction. You are correct that our original claim was an oversimplification. Our statement was intended to refer to the classical jigsaw formulation from early self-supervised learning, which, while foundational, was often surpassed in large-scale pre-training by later mainstream approaches. As the reviewer notes, more recent jigsaw-style SSL variants can perform on par with contrastive learning and masked image modeling. We have revised the text to reflect this problem and added discussion of these recent jigsaw-style methods in the relate work section.
>
> > **Q3: Claims about predominantly text-centric post-training paradigms.**
>
> **A3:**  Thanks for pointing this out! We have modified the claims and added the relevant discussion in the related work section. We acknowledge a growing line of thinking-with-image approaches that better extract/inspect visual information via certain operations like crop and zoom-in to acquire clearer views when the raw input is insufficient.
>
> Our Visual Jigsaw framework addresses a different, but we believe complementary, goal: enhancing intrinsic visual perception. Rather than teaching the model to acquire new visual data, our objective is to fundamentally improve its ability to interpret and reason about the visual information it is already given.
> We see these two approaches as highly synergistic:
>
> *   A model with a stronger intrinsic understanding of spatial relationships and object composition (honed by Visual Jigsaw) will be better equipped to make intelligent decisions about *when* and *where* to use tools like zoom-in or crop.
> *   Conversely, the ability to actively gather information can provide clearer inputs that make the intrinsic reasoning process more effective.
>
> We plan to explore their combination in future work.

---

> ### Author Response · Authors · 2025-11-18
> **Response to reviewer kYxc (Part 2)**
>
> > **Q4: Quantative Examples.**
>
> **A4:** Thanks for the great suggestion. To provide a more in-depth qualitative analysis of our model's reasoning process and a clearer mechanistic explanation of *why* the Visual Jigsaw task enhances visual understanding. We address this in two parts.
>
> ---
>
> #### **Part 1: Qualitative Analysis of Textual Reasoning**
>
> To illustrate the model's problem-solving process, we provide a representative example of its "thinking traces" (`<think>...</think>`) when faced with the image task. More examples with the corresponding visual information are in the revised paper (Section 4.3 and Appendix A.1)
>
> *   **Image Jigsaw Example (Snowy Scene):**
>     *   `<think>` I am mentally reassembling the image based on the spatial relationships and context of the visible elements in each tile. Tile 8 is positioned at the top-left corner of the image, showing part of a building with a partial view of a street or urban environment, likely indicating the upper-left side of a scene with buildings and possibly residential or commercial structures. Tile 1 occupies the top-right portion of the image, showing a continuation of the urban environment with trees and parts of a building facade, consistent with a street or plaza area. Tile 7 appears to be situated on the left side, showing people from behind, interacting near what seems like a market or street vendor setup, with boxes and goods visible, suggesting this could be a marketplace or public gathering area. Tile 6, located centrally towards the right, shows a continuation of the market activity with a focus on goods being sold, including produce and signage, consistent with a busy street market environment. Tile 3, positioned on the lower right, shows a continuation of the street scene with a vehicle and pavement, providing a sense of depth and perspective to the lower right-hand side of the image, completing the street-level view. By integrating these observations, I deduce the spatial organization of the image. `</think>`
>     *   **Analysis:** The model identifies semantic objects (trees, sky, a person), infers spatial context (horizon line, pathway edges), and uses logical deduction to place the tiles into a coherent global scene.
>
>
> While these thinking traces provide valuable insight, we argue that they are a scaffold for learning, not the entirety of it. The ultimate goal is to improve the model's **intrinsic and direct visual comprehension**, which often operates pre-verbally. And when evaluating downstream benchmarks, we require the model to provide **direct answers**, ensuring we are measuring its intrinsic visual capabilities.
>
> ---
>
> #### **Part 2: The Underlying Mechanism: How Jigsaw Deconstructs and Teaches Visual Skills**
>
> The Visual Jigsaw task is effective because it forces the model to develop different fundamental, vision-centric sub-skills. We break down these requisite skills (based on the analysis of the thinking traces and also human priors about this task) and validate them by improvements on certain downstream benchmarks. We choose benchmarks or subsets of them that **directly measure certain visual-centric ability.**
>
> We take the image jigsaw as an example. Solving a 3x3 image jigsaw puzzle implicitly requires and thus teaches the following:
>
> *   **Fine-Grained Grounding and Recognition:** The model must accurately ground on and reference the small image tiles and read their local content. This directly enhances its ability to ground specific targets and understand the direct attributes.
>     *   **Evidence:** We see improvement on the V*-attribute subset (from 78.26 to 81.73) , which probes accurate grounding and simple attribute understanding.
>
> *   **Spatial and Relational Reasoning:** The model must understand concepts like "above" and "to the left of" for objects in the scene to assemble the scene.
>     * **Evidence:** This skill is directly reflected in improved scores (77.68 to 80.36) on the **VSR** benchmark, which tests the understanding of spatial relationships between objects.
>
> *   **Depth and Geometry Understanding:** Correct ordering often hinges on depth cues (horizon/vanishing lines, size/occlusion).
>     *   **Evidence:** Post-training on image jigsaw leads to a marked improvement (54.45 to 60.35) on the monocular depth estimation **DA-2K** benchmark.
>
> * **Visual correspondence & boundary consistency:** Reassembling requires matching edges, textures, and continuation of structures across tiles.
>     *   **Evidence:** This maps to improvements on **BLINK-visual correspondence** subset (from 45.34 to 54.06).
>
> By gradually mastering the jigsaw task, the model inherently sharpens these fundamental skills, leading to more robust overall visual understanding. And it is similar for the video and 3D cases.

---

> ### Comment · Reviewer_kYxc · 2025-11-21
>
> I really appreciate the authors' rebuttal, especially the updated "multi-task" experiments. I have checked the revised version, and my concerns regarding inappropriate claims have been addressed.
>
> However, both Q1 and Q4 are *partially* addressed. Specifically,
> - Regarding Q1, it is encouraged to demonstrate the effectiveness of "cross-domain" post-training, i.e., testing the image checkpoint on 3D and vice versa. This would help to strengthen the generalization capabilities of the proposed post-training method.
> - Regarding Q4, I notice the authors require the model to provide direct answers when evaluating downstream benchmarks, which is quite strange for a reasoning-oriented model. Therefore, a comparison between the thinking and non-thinking modes at test time is encouraged.

---

> > ### Author Response · Authors · 2025-11-21
> > **Response to Reviewer kYxc (Round 2)**
> >
> > We thank the reviewer for the continued engagement and the positive assessment of our rebuttal regarding the claims. We are happy to address the remaining questions regarding cross-domain generalization and the inference mode (thinking vs. non-thinking).
> >
> > > **Q1: Cross-domain post-training effectiveness (Image checkpoint on 3D and vice versa).**
> >
> > **A1:** This is an excellent suggestion to verify the generalization capabilities of our method. We have conducted the requested cross-domain evaluation.
> >
> > To ensure a rigorous assessment of "cross-domain" generalization, we first adjusted our Image Benchmark Avg metric. Since our original image benchmark suite included 3D-related tasks (specifically DA-2K and OmniSpatial), we excluded these from the "Image Benchmark Avg" reported below.
> >
> > | Model | Image Benchmark Avg (Strict 2D*) | 3D Benchmark Avg |
> > | :--- | :---: | :---: |
> > | Qwen2.5-VL-7B (Baseline) | 60.79 | 45.09 |
> > | Image Jigsaw | 63.82 | 48.61 |
> > | 3D Jigsaw | 61.52 | 50.78 |
> >
> > *\*Note: The strict 2D average excludes DA-2K and OmniSpatial to prevent task overlap.*
> >
> > **Observations & Analysis:**
> > As shown in the table, we observe clear bidirectional transfer: the Image Jigsaw model improves on 3D benchmarks, and the 3D Jigsaw model improves on strict 2D image benchmarks. We attribute this to the **overlap between the fundamental capabilities required to solve a specific Jigsaw task and those required by the cross-domain benchmarks**:
> >
> > 1.  **From 3D Jigsaw $\rightarrow$ 2D Benchmarks:** To successfully solve the 3D Jigsaw task (ranking points from closest to farthest), the model also needs to attend to specific objects and capture fine-grained details. This enhances its fine-grained grounding and object understanding capabilities, which are critical for and directly transferable to general 2D image tasks.
> > 2.  **From 2D Image Jigsaw $\rightarrow$ 3D Benchmarks:** Conversely, solving the 2D Image Jigsaw requires the model to infer the global scene layout based on local patches. This implicitly requires judging spatial relationships and relative depth cues (e.g., occlusion, perspective). These spatial and depth-aware capabilities generalize well to 3D tasks.
> >
> > Thus, even though the training domains differ, the specific visual skills cultivated by our Jigsaw tasks are foundational and generalize effectively to other domains.
> >
> >
> > > **Q4: Comparison between thinking and non-thinking modes at test time.**
> >
> > **A4:** We appreciate this suggestion. We would like to clarify our rationale for choosing direct evaluation for the main experiments, while also highlighting new results that address your preference for reasoning-oriented evaluation.
> >
> > **1. Rationale for Direct Evaluation on Qwen2.5-VL Baseline:**
> > Our choice of direct evaluation was driven by the specific nature of the selected vision-centric benchmarks and the behavior of the base model:
> > *   **Perceptual vs. Reasoning Nature:** Most of the selected benchmarks focus on direct visual perception and understanding which does not require complex, multi-step text-based reasoning.
> > *   **Instability of "Thinking" on Perceptual Tasks:** We empirically observed that forcing the Qwen2.5-VL baseline to "think" before answering on these specific benchmarks often led to **performance degradation and high variance**. This is likely because the cot process brings additional hallucinations. Robust CoT performance typically requires extensive SFT and RL on diverse general data, which the Qwen2.5-VL 7B base model may not fully possess for these specific formats.
> > *   **Fairness:** To eliminate these interference factors and fairly evaluate the intrinsic vision understanding capability gained from our pre-training, we thus opted for direct output.
> >
> > The table below illustrates the performance when forcing the baseline model to "think" on some of the vision-centric tasks:
> >
> > | Benchmark | Baseline (Direct) | Baseline (Thinking) | Image Jigsaw (Direct) | Image Jigsaw (Thinking)|
> > | :--- | :---: | :---: | :---: | :---: |
> > | MMStar-fine | 59.75 | 50.26 | 65.81 | 59.59 |
> > | MMBench | 83.33 | 78.35 | 84.45 | 80.67 |
> > | V* | 76.96 | 72.77 | 80.62 | 82.19 |
> > | VSR | 77.68 | 68.66 | 80.36 | 80.82 |
> > | MME-Realworld-Lite | 43.41 | 44.09 | 45.96 | 49.30 |
> >
> > **2. Evaluation on Stronger Reasoning Baseline (MiMo-VL):**
> > However, we fully agree with your point regarding reasoning-oriented models. In our revised paper (Section 4.3 Table 4), we have introduced experiments with a stronger base model, MiMo-VL, which has undergone more extensive training to support stable CoT reasoning before answering.
> > *   For the MiMo-VL experiments, we **adopted the "Thinking" mode for all evaluations** as you suggested.
> > *   The results confirm that our Visual Jigsaw post-training also provides consistent gains in this setting, proving the method's effectiveness across different base models and inference modes.

---

### Official Review · Reviewer_uvM4 · 2025-10-24

**Soundness:** 3
**Presentation:** 4
**Contribution:** 3
**Rating:** 6
**Confidence:** 5

**Summary:**

This paper introduces a novel post-training task called Visual Jigsaw, which encourages MLLMs to learn the inherent order within visual inputs. An RL-based supervision strategy is employed to train the model. Experiments on image, video, and 3D benchmarks demonstrate that the proposed method significantly improves performance.

**Strengths:**

1.	The paper presents an effective post-training framework for enhancing MLLMs.

2.	The idea of leveraging a self-supervised objective to strengthen the visual understanding capability of MLLMs is novel and promising.

3.	The proposed approach achieves strong performance across diverse benchmarks, including image, video, and 3D tasks.

4.	The manuscript is well structured, clearly written, and easy to follow.

**Weaknesses:**

1.	The Visual Jigsaw task employs a relatively simple supervised objective. Although the experimental results across multiple benchmarks are promising, the underlying mechanism of how this form of supervision contributes to model improvement remains unclear. It would strengthen the paper to include an analysis of what is learned during post-training and how this supervision enhances visual comprehension.

2.	The experiments are conducted on only a single MLLM. Including additional baseline models would help demonstrate the generalizability and robustness of the proposed Visual Jigsaw framework.

3.	The Visual Jigsaw post-training is performed using specific individual datasets rather than the original MLLM training data. It would be helpful to clarify the rationale behind this choice and provide guidelines for selecting appropriate post-training data.

**Questions:**

See weaknesses.

---

> ### Author Response · Authors · 2025-11-18
> **Response to reviewer uvM4 (Part 1)**
>
> We are very grateful to the reviewer for the supportive and insightful comments. We address your valuable points in our response below.
>
> ---
>
> > **Q1: Underlying mechanism of how training with visual jigsaw contributes to model improvement.**
>
> **A1:** Thanks for the great suggestion. To provide a more in-depth qualitative analysis of our model's reasoning process and a clearer mechanistic explanation of *why* the Visual Jigsaw task enhances visual understanding. We address this in two parts.
>
> ---
>
> #### **Part 1: Qualitative Analysis of Textual Reasoning**
>
> To illustrate the model's problem-solving process, we provide a representative example of its "thinking traces" (`<think>...</think>`) when faced with the image task. More examples with the corresponding visual information are in the revised paper (Section 4.3 and Appendix A.1)
>
> *   **Image Jigsaw Example (Snowy Scene):**
>     *   `<think>` I am mentally reassembling the image based on the spatial relationships and context of the visible elements in each tile. Tile 8 is positioned at the top-left corner of the image, showing part of a building with a partial view of a street or urban environment, likely indicating the upper-left side of a scene with buildings and possibly residential or commercial structures. Tile 1 occupies the top-right portion of the image, showing a continuation of the urban environment with trees and parts of a building facade, consistent with a street or plaza area. Tile 7 appears to be situated on the left side, showing people from behind, interacting near what seems like a market or street vendor setup, with boxes and goods visible, suggesting this could be a marketplace or public gathering area. Tile 6, located centrally towards the right, shows a continuation of the market activity with a focus on goods being sold, including produce and signage, consistent with a busy street market environment. Tile 3, positioned on the lower right, shows a continuation of the street scene with a vehicle and pavement, providing a sense of depth and perspective to the lower right-hand side of the image, completing the street-level view. By integrating these observations, I deduce the spatial organization of the image. `</think>`
>     *   **Analysis:** The model identifies semantic objects (trees, sky, a person), infers spatial context (horizon line, pathway edges), and uses logical deduction to place the tiles into a coherent global scene.
>
>
> While these thinking traces provide valuable insight, we argue that they are a scaffold for learning, not the entirety of it. The ultimate goal is to improve the model's **intrinsic and direct visual comprehension**, which often operates pre-verbally. And when evaluating downstream benchmarks, we require the model to provide **direct answers**, ensuring we are measuring its intrinsic visual capabilities.
>
> ---
>
> #### **Part 2: The Underlying Mechanism: How Jigsaw Deconstructs and Teaches Visual Skills**
>
> The Visual Jigsaw task is effective because it forces the model to develop different fundamental, vision-centric sub-skills. We break down these requisite skills (based on the analysis of the thinking traces and also human priors about this task) and validate them by improvements on certain downstream benchmarks. We choose benchmarks or subsets of them that **directly measure certain visual-centric ability.**
>
> We take the image jigsaw as an example. Solving a 3x3 image jigsaw puzzle implicitly requires and thus teaches the following:
>
> *   **Fine-Grained Grounding and Recognition:** The model must accurately ground on and reference the small image tiles and read their local content. This directly enhances its ability to ground specific targets and understand the direct attributes.
>     *   **Evidence:** We see improvement on the V*-attribute subset (from 78.26 to 81.73) , which probes accurate grounding and simple attribute understanding.
>
> *   **Spatial and Relational Reasoning:** The model must understand concepts like "above" and "to the left of" for objects in the scene to assemble the scene.
>     * **Evidence:** This skill is directly reflected in improved scores (77.68 to 80.36) on the **VSR** benchmark, which tests the understanding of spatial relationships between objects.
>
> *   **Depth and Geometry Understanding:** Correct ordering often hinges on depth cues (horizon/vanishing lines, size/occlusion).
>     *   **Evidence:** Post-training on image jigsaw leads to a marked improvement (54.45 to 60.35) on the monocular depth estimation **DA-2K** benchmark.
>
> * **Visual correspondence & boundary consistency:** Reassembling requires matching edges, textures, and continuation of structures across tiles.
>     *   **Evidence:** This maps to improvements on **BLINK-visual correspondence** subset (from 45.34 to 54.06).
>
> By gradually mastering the jigsaw task, the model inherently sharpens these fundamental skills, leading to more robust overall visual understanding. And it is similar for the video and 3D cases.

---

> ### Author Response · Authors · 2025-11-18
> **Response to reviewer uvM4 (Part 2)**
>
> > **Q2: Additional baseline models.**
>
> **A2:** Thanks for the suggestion. We have additionally applied our proposed visual jigsaw task on MiMo-VL-7B-SFT-2508 with the results shown below, where visual jigsaw yields consistent gains over the baseline.
>
> | Model           | Image Benchmark Avg | Video Benchmark Avg | 3D Benchmark Avg |
> |------------------|-------|-------|-------|
> | MiMo-VL-7B-SFT  | 63.77   | 51.84  | 50.67 |
> | + Visual Jigsaw  | 65.14   | 54.47  | 52.91 |
>
>
> > **Q3: Rationale behind using specific individual datasets.**
>
> **A3:** This is a good question regarding our data selection strategy. First, the full pre-training data mixture for the base MLLM is proprietary, making it impossible to use for reproducible research. Second, and more fundamentally, our self-supervised approach only requires the raw visual data (images, videos), not the complex text-image annotations from the original pre-training set. We therefore opted for widely-used, public benchmarks, COCO (images), LLaVA-Video (videos), and ScanNet (3D), which are themselves the sources for many MLLM instruction-tuning datasets. This ensures our methodology is transparent, reproducible, and built upon standard, high-quality data.
>
> We agree that how to select post-training data is important. While an exhaustive study on data selection is beyond our current scope, we can offer the following heuristics derived from our experience:
>
> *   **Image Data:** Datasets with proper resolution and rich structural complexity are ideal. A key heuristic is to filter out images or patch pairs with low information content or high visual similarity (e.g., large patches of uniform sky) to avoid creating ambiguous or ill-posed puzzles.
> *   **Video Data:** The most effective videos contain clear causal or temporal progression. We recommend prioritizing footage with continuous actions and clear motion over collections of static scenes or rapidly edited clips.
> *   **3D Data:** For our depth-ordering task, data quality is paramount. The best sources are datasets with accurate depth maps and salient geometric structures that provide unambiguous depth cues.

---

### Official Review · Reviewer_znHd · 2025-10-28

**Soundness:** 3
**Presentation:** 3
**Contribution:** 3
**Rating:** 4
**Confidence:** 3

**Summary:**

This paper proposes Visual Jigsaw , a self-supervised post-training framework designed to enhance the intrinsic visual understanding of MLLMs. The core idea is to formulate a visual ordering task, the model is presented with shuffled parts of a visual input and must generate a text-based permutation to reconstruct the original order. This approach cleverly leverages RLVR, specifically using GRPO, to optimize the model. The authors instantiate this method across images, videos, and 3D data . Extensive experiments on a wide range of benchmarks show that this post-training stage significantly improves fine-grained perception, temporal reasoning, and 3D spatial understanding, demonstrating the method's generality and effectiveness.

**Strengths:**

The paper presents an elegant and effective idea. It repurposes a classic self-supervised task (jigsaw puzzles) into a post-training stage for modern MLLMs. The framework is simple, requires no architectural modifications or extra generative modules, and is broadly applicable.

**Weaknesses:**

The improvement effect of some benchmarks is not significant.

**Questions:**

1. Have you tried expanding on the harder puzzle tasks (like 4x4)?
2. Given that some improvements are modest, the results would be more credible if variance  of evaluation are reported.

---

> ### Author Response · Authors · 2025-11-18
> **Response to reviewer znHd**
>
> We thank the reviewer for the feedback. We are pleased that you found our paper to present an elegant and effective idea. We address each point raised below.
>
> ---
>
> > **Q1: The improvement effect of some benchmarks is not significant.**
>
> **A1:** Our evaluation was designed to rigorously test for broad and consistent enhancement of vision-centric skills, rather than optimizing for performance on a single leaderboard. Therefore, we believe the most telling metric is the average performance gain across our comprehensive suite of evaluations, which includes 13 image, 13 video (across three settings), and 8 diverse 3D benchmarks.
> Across this suite, Visual Jigsaw achieves a mean improvement of +3.20% on image benchmarks, +2.20% on video benchmarks, and +5.69% on 3D benchmarks. We consider these average gains to be highly significant and impactful, particularly as they are achieved on top of an already strong base model.
>
> The smaller margins on certain individual benchmarks are expected in such a wide-ranging evaluation and can be attributed to factors like task saturation at the current model scale or a lesser dependence on the specific structural and relational skills our method enhances. However, the consistently positive trend across the vast majority of tasks, combined with the substantive average gains across all three modalities, provides robust evidence that our self-supervised task broadly and effectively strengthens the model's core visual understanding capabilities.
>
> > **Q2: Harder puzzle tasks.**
>
> **A2:** Thanks for the suggestion. We have additionally tried **harder** setups.
> For Image Jigsaw, we tested a 4×4 grid. We found that the model failed to learn this task effectively, with jigsaw accuracy remaining low. This happens even when we do cross-configuration adaptation by first training on 3×3 jigsaw task. We attribute this to three key factors:
>
> (1) **Information Scarcity**: At the typical resolution of our COCO training data (~640×480), 4×4 tiles are often too small and fragmented to contain sufficient semantic information for re-assembly.
>
> (2) **Semantic Ambiguity**: Small tiles from uniform areas (e.g., sky, walls) become perceptually indistinguishable, making the ordering task ill-posed.
>
> (3) **Combinatorial Explosion**. The permutation space expands from 9! to 16!, a significant increase in complexity for the model.
>
> This suggests that scaling to harder configurations is not merely a matter of increasing grid size, but will likely require a combination of higher-resolution training data, tile-diversity constraints, and curriculum learning strategies.
>
> For video, we added **8-clip** and **4-clip + 2×2 spatial** variants (evaluated under the 16-frame setting).
>
> As shown in the table, **8-clip** performs on par with 6-clip; this is because many LLaVA-Video samples are ~30 s, so splitting into 8 clips makes segments short and sometimes ambiguous. We plan to revisit 8-clip on longer videos in the future. The **4-clip + 2×2 spatial** configuration outperforms temporal-only 4-clip, indicating the value of adding spatial structure to the temporal ordering signal. Further exploring adaptive clip lengths and richer spatio-temporal grids is a promising direction which we leave for future work.
>
> > **Q3: variance of evaluation.**
>
> **A3:** This is an excellent point. We agree that reporting variance is crucial for demonstrating the robustness of our findings. First, we clarify that our evaluation protocol eliminates randomness at test time by using deterministic decoding (temperature = 0). The more important source of variance, therefore, is the stability of the training process itself. To assess this, we have already performed three independent training runs for our Image Jigsaw experiment. The results show an average score of 61.96 with a low standard deviation of 0.07, confirming that the gains are robust and reproducible across runs.
>
> We will extend this rigorous validation protocol to all modalities. We will repeat each key experiment with multiple random seeds and report mean ± standard deviation for all results, providing a comprehensive assessment of the variability and further strengthening the reliability of our claims.

---

### Official Review · Reviewer_Dhxc · 2025-11-01

**Soundness:** 3
**Presentation:** 3
**Contribution:** 2
**Rating:** 4
**Confidence:** 4

**Summary:**

This paper introduces Visual Jigsaw, a self-supervised post-training framework designed to enhance vision-centric understanding in MLLMs. The method formulates a general visual ordering task, partitioning and shuffling images, videos, or 3D inputs and requires the model to predict the correct permutation via natural-language output without modifying architecture or adding visual generators. Experiments on Qwen2.5-VL show improvements across fine-grained perception, temporal reasoning, and 3D spatial understanding benchmarks.

**Strengths:**

- The jigsaw ordering task provides a simple yet effective way to reinforce visual perception in MLLMs, avoiding generative components.

- Applying the post-training principle to images, videos, and 3D data convincingly demonstrates versatility.

- Evaluation on various benchmarks covering visual perception, temporal reasoning, and 3D geometry provide good empirical support.

- SFT vs RL comparison, jigsaw-difficulty analysis, and transfer to reasoning models (ThinkLite-VL) strengthen the claims.

- The paper is well written and figures are intuitive.

**Weaknesses:**

- The paper has limited conceptual novelty. The idea extends classical self-supervised “jigsaw” pretexts (Noroozi & Favaro 2016) into RL post-training. The novelty mainly lies in adapting it to MLLM post-training rather than the task itself.

- Missing related works of reconstruction-based methods in MLLMs and discussion in paper: Recent works like X-Former[1] explicitly combines contrastive and masked-reconstruction objectives with frozen encoder and decoders to improve visual understanding during pre-training with less data. In contrast, Visual Jigsaw uses structural ordering without dense pixel reconstruction. The paper should discuss this design trade-off in detail, why post-training with ordering might achieve similar benefits, and whether the two paradigms (reconstruction vs ordering) could be complementary.

- The paper omits discussion on training overhead for Jigsaw compared to SFT.

- Missing comparison to SoTA for image and video benchmarks.

- The paper could further analyze reward sensitivity and scalibility (beyond 3×3 grid/6-clips).

- All results rely on Qwen2.5-VL; would be good to show on other model architectures like LLaVA/Blip to confirm generality

- Missing qualitative analysis. Visual reasoning traces (< think > outputs) could better substantiate the claim of improved visual understanding.



[1] X-Former: Unifying Contrastive and Reconstruction Learning for MLLMs. Sirnam Swetha, Jinyu Yang, Tal Neiman, Mamshad Nayeem Rizve, Son Tran, Benjamin Yao, Trishul Chilimbi, Mubarak Shah. ECCV 2024

**Questions:**

- How sensitive are the improvements to the number of jigsaw pieces (K)? Does performance saturate or degrade with grid size (4×4, 5×5) grids or > 6 video clips?

- The reward combines partial correctness with a discount factor γ = 0.2. How sensitive are results to this value? Did the authors explore other reward formulations ?

- What is the total cost of post-training ?

---

> ### Author Response · Authors · 2025-11-18
> **Response to reviewer Dhxc (Part 1)**
>
> We thank the reviewer for the detailed feedback and constructive suggestions. Below, we address each point raised.
>
> ---
> > **Q1: Limited conceptual novelty.**
>
> **A1:** We argue the core innovation of our work lies in demonstrating how a classic, vision-centric self-supervised task can be successfully and non-trivially adapted for the modern MLLM post-training paradigm. In contrast to conventional post-training methods that rely on VQA-style data and its inherent language priors, our work reintroduces an objective that operates purely on visual signals. We contend that successfully formulating this adaptation and demonstrating its effectiveness across images, videos, and 3D, provides a novel and practical path for enhancing the intrinsic visual understanding of MLLMs.
> We believe that re-focusing on vision-centric self-supervised signals is crucial for the advancement of MLLMs, and we hope our work inspires more research into low-language-prior self-supervised or weakly-supervised objectives for multimodal models. The emergence of **very recent works [1, 2, 3] also exploring vision-centric self-supervised MLLM training** further underscores the timeliness and importance of this research direction.
>
> [1] Liu, Yuhong, et al. "Spatial-SSRL: Enhancing Spatial Understanding via Self-Supervised Reinforcement Learning." arXiv preprint arXiv:2510.27606 (2025).
> [2] He, Zefeng, et al. "VideoSSR: Video Self-Supervised Reinforcement Learning." arXiv preprint arXiv:2511.06281 (2025).
> [3] Fang, Bo, et al. "ViSS-R1: Self-Supervised Reinforcement Video Reasoning." arXiv preprint arXiv:2511.13054 (2025).
>
> > **Q2: Missing related works of reconstruction-based methods like X-Former in MLLMs.**
>
> **A2:** Thank you for this valuable suggestion. We have revised Section 2.2 to include X-Former and expand on this important comparison. We clarify that X-Former's reconstruction objective is designed to strengthen the **connector module** to extract better visual features *for* the LLM. The reconstruction is handled by a light-weight connector and **does not pass through the LLM**. In contrast, Visual Jigsaw is designed to improve the visual understanding of the **entire MLLM**. It requires the LLM to directly interpret, reason about, and process visual features to generate the correct permutation in natural language.
>
> A closer comparison is ROSS [1] (already discussed in our paper), which represents the dense visual reconstruction paradigm in MLLM. Its approach needs additional generative modules and has mainly been shown when training from scratch; whether dense reconstruction can further improve top-performing models (e.g., Qwen2.5-VL) and how to do this remains an open question. Dense reconstruction also biases learning toward modeling vision tokens themselves, while our ordering signal targets structural/relational cues and information extraction from visual signals. Moreover, for MLLMs, it is still unclear whether dense-level visual reconstruction is necessary, as some UMM studies suggest that adding generative tasks can even hurt understanding abilities.
>
> Finally, **the two approaches are not conflicting but complementary** as we can **directly incorporate the visual reconstruction objective from ROSS during our visual jigsaw training**. We will also explore their combination in future experiments.
>
> [1] Wang, Haochen, et al. "Reconstructive visual instruction tuning." ICLR 2025.
>
> > **Q3: Training overhead of RL and SFT. Total cost of post-training**
>
> **A3:** Thank you for raising this important point. We have added a detailed breakdown of the training costs to Section 4.1 and further discuss the comparison in Section 4.3 in the revised paper. For the image jigsaw task, SFT required 58 H100 GPU-hours, while RL required 840 GPU-hours. The RL costs for the video and 3D jigsaw tasks were 1600 and 310 GPU-hours, respectively. The higher cost of RL is expected, as the on-policy training process requires generating multiple rollouts for each sample to estimate rewards and update the policy.
>
> Crucially, this additional computational investment is justified by the substantially improved performance. As we demonstrate in Section 4.3, RL training leads to significantly stronger and more transferable generalization compared to SFT. In essence, the extra compute is spent on achieving robust generalization rather than simple task memorization.
>
> > **Q4: Comparison to SoTA for image and video benchmarks.**
>
> **A4:** Thank you for this suggestion. To provide a clearer context for our results, we will add SoTA comparisons to the main tables for the relevant benchmarks in the next version.

---

> ### Author Response · Authors · 2025-11-18
> **Response to reviewer Dhxc (Part 2)**
>
> > **Q5: Reward sensitivity (choice of γ)**
>
> **A5:** Thank you for this question about a key hyperparameter in our reward design. Our primary motivation for designing a graded reward with partial correctness was to provide sufficient learning signals in the early stages of training. To empirically test the sensitivity to the discount factor γ, we conducted an ablation study on the image jigsaw task. The results are as follows:
>
> | Model          | Image Benchmark Avg |
> |------------------|-------|
> | Baseline  |  58.91 |
> | γ = 0  |  Fail to learn |
> | γ = 0.2  |  62.06 |
> | γ = 0.5  |  61.83 |
> | γ = 1.0  |  61.89 |
>
> We can see that a graded reward is essential for learning; a purely binary reward (γ = 0) makes the model fail to overcome the task's complexity. Second, once a partial credit signal is introduced (γ > 0), the model's final performance is robust to the specific value of the discount factor. The similar results for γ = 0.2, 0.5, and 1.0 demonstrate that the key design choice is the presence of a graded reward, not its precise magnitude.
>
> > **Q6: Different Jigsaw Configurations (beyond 3×3 grid/6-clips)**
>
> **A6:**
> This is an excellent question. We agree that task difficulty is a critical design choice. As shown in Section 4.3, easier tasks (2×2 grid; 4-clip video) yield smaller gains. To investigate the upper bound of difficulty, we conducted new experiments with harder configurations.
>
> For Image Jigsaw, we tested a 4×4 grid. We found that the model failed to learn this task effectively, with jigsaw accuracy remaining low. This happens even when we do cross-configuration adaptation by first training on 3×3 jigsaw task. We attribute this to three key factors:
>
> (1) **Information Scarcity**: At the typical resolution of our COCO training data (~640×480), 4×4 tiles are often too small and fragmented to contain sufficient semantic information for re-assembly.
>
> (2) **Semantic Ambiguity**: Small tiles from uniform areas (e.g., sky, walls) become perceptually indistinguishable, making the ordering task ill-posed.
>
> (3) **Combinatorial Explosion**. The permutation space expands from 9! to 16!, a significant increase in complexity for the model.
>
> This suggests that scaling to harder configurations is not merely a matter of increasing grid size, but will likely require a combination of higher-resolution training data, tile-diversity constraints, and curriculum learning strategies. We will further explore this in the future.
>
> For video, we added **8-clip** and **4-clip + 2×2 spatial** variants.
>
> | Model          | Video Benchmark Avg (16-frame) |
> |------------------|-------|
> | 4-clip  |  45.05 |
> | 6-clip  |  46.18 |
> | 8-clip  |  46.24 |
> | 4-clip + 2×2 spatial|  45.79 |
>
> As shown in the table, **8-clip** does not bring notable improvement over 6-clip; this is probably because many LLaVA-Video samples are ~30 s, so splitting into 8 clips makes segments short and sometimes ambiguous. We plan to revisit 8-clip on longer videos in the future. The **4-clip + 2×2 spatial** configuration outperforms temporal-only 4-clip, indicating the value of adding spatial structure to the temporal ordering signal. Further exploring adaptive clip lengths and richer spatio-temporal grids is a promising direction which we leave for future works.
>
>
> > **Q7: Other model architectures to confirm generality.**
>
> **A7:** Thank you for this important question regarding the generality of our method. Our initial choice of Qwen2.5-VL was motivated by several factors critical for a rigorous study: 1) A strong baseline is necessary for the RL training to receive meaningful learning signals from its on-policy rollouts. 2) The model's wide adoption in research community and 3) the mature tooling ecosystem (e.g., Verl for RL, vLLM for inference) ensure that our findings are both relevant and easily reproducible by the community.
>
> Importantly, current high-performing MLLMs are converging architecturally (e.g., lightweight MLP connectors and native-resolution inputs in Qwen2.5/3-VL, MiMo-VL, Keye-VL, GLM-4.5V, Kimi-VL, LLaVA-OV-1.5), and our designed task is architecture-agnostic, as it does not rely on special decoders or connector designs.
>
> To show the effectiveness of our method on other base models, we have added experiments on MiMo-VL-7B-SFT-2508, where Visual Jigsaw also yields consistent gains over the baseline.
>
> | Model           | Image Benchmark Avg | Video Benchmark Avg | 3D Benchmark Avg |
> |------------------|-------|-------|-------|
> | MiMo-VL-7B-SFT  | 63.77   | 51.84  | 50.67 |
> | + Visual Jigsaw  | 65.14   | 54.47  | 52.91 |

---

> ### Author Response · Authors · 2025-11-18
> **Response to reviewer Dhxc (Part 3)**
>
> > **Q8: Qualitative analysis.**
>
> **A8:** To illustrate the model's problem-solving process, we provide examples of its "thinking traces" (`<think>...</think>`) when faced with the image, video, and 3D jigsaw tasks. We have also added more examples with the corresponding visual information in our revised paper (Section 4.3 and Appendix A.1)
>
> *   **Image Jigsaw Example (Snowy Scene):**
>     *   `<think>` I am mentally reassembling the image based on the spatial relationships and context of the visible elements in each tile. Tile 8 is positioned at the top-left corner of the image, showing part of a building with a partial view of a street or urban environment, likely indicating the upper-left side of a scene with buildings and possibly residential or commercial structures. Tile 1 occupies the top-right portion of the image, showing a continuation of the urban environment with trees and parts of a building facade, consistent with a street or plaza area. Tile 7 appears to be situated on the left side, showing people from behind, interacting near what seems like a market or street vendor setup, with boxes and goods visible, suggesting this could be a marketplace or public gathering area. Tile 6, located centrally towards the right, shows a continuation of the market activity with a focus on goods being sold, including produce and signage, consistent with a busy street market environment. Tile 3, positioned on the lower right, shows a continuation of the street scene with a vehicle and pavement, providing a sense of depth and perspective to the lower right-hand side of the image, completing the street-level view. By integrating these observations, I deduce the spatial organization of the image. `</think>`
>     *   **Analysis:** The model identifies semantic objects (trees, sky, a person), infers spatial context (horizon line, pathway edges), and uses logical deduction to place the tiles into a coherent global scene.
>
> *   **Video Jigsaw Example (Drawing Hello Kitty):**
>     *   `<think>` To determine the correct order, we analyze the logical progression of drawing Hello Kitty:  \n\n1. **Clip 2** begins with a hand holding a pen, then starts drawing the initial U - shape for the head.  \n2. **Clip 4** builds on Clip 2 by adding the bow to the head.  \n3. **Clip 6** continues by drawing the eyes, nose, and whiskers on the head.  \n4. **Clip 3** moves to the body, drawing the arms and starting the lower part of the body.  \n5. **Clip 5** further develops the body, adding more details to the limbs.  \n6. **Clip 1** completes the body with additional details like stripes. `</think>`
>     *   **Analysis:** The model must grasp the temporal logic and causal sequence of an action. It correctly identifies the intermediate stages of a process, demonstrating an understanding of the "arrow of time."
>
> *   **3D Jigsaw Example (Office Scene):**
>     *   `<think>` Let's analyze this step-by-step:\n\n1. **Point 6**: It's right in the foreground on the table.\n2. **Point 3**: It's a bit further back on the table.\n3. **Point 1**: It's on the table but a little further back than Point 3.\n4. **Point 2**: It's on the wall in the background.\n5. **Point 5**: It's also on the wall but a bit higher up.\n6. **Point 4**: It's on the wall to the left of Point 2. `</think>`
>     *   **Analysis:** The model correctly identifies distinct depth layers (table vs. wall) and then reasons about the relative positions of points within each layer based on perspective cues (e.g., lower on the table is closer). This demonstrates an understanding of how 2D images represent 3D space.
>
> While these thinking traces provide valuable insight, we argue that they are a scaffold for learning, not the entirety of it. The ultimate goal is to improve the model's *intrinsic and direct visual comprehension*, which often operates pre-verbally.

---

### Public Comment · ~Junyi_Zhu1 · 2025-11-17
**Link to Jigsaw-R1**

Thanks for sharing this interesting work! I enjoyed reading it and I’m glad to see continuous interest in rule-based RL and structured pretext tasks in multimodal learning.

I’m one of the authors of *Jigsaw-R1: A Study of Rule-based Visual Reinforcement Learning with Jigsaw Puzzles* (Wang et al.; preprint May 2025; accepted to TMLR Oct 2025), which you already cite as related work. Thank you. Our paper also introduces jigsaw puzzles as a rule-based visual RL pretext task for post-training multimodal LLMs, and studies their impact on generalization and reasoning. To the best of our knowledge, Jigsaw-R1 was the first to use jigsaw puzzles in this specific “MLLM + rule-based RL post-training” setting.

From a reader’s perspective, the connection between the two works could be made a bit more explicit. Since both papers study jigsaw-based RL for MLLMs, some experimental comparison on overlapping settings, would help the community understand the relative strengths of each approach.

This is meant purely as a constructive suggestion. I’m very happy to see follow-up work in this direction, and I think your extensions to video and 3D, along with the graded reward and more complex ordering tasks, provide useful perspective on how structured visual pretext tasks can improve MLLM perception under rule-based RL.

---

> ### Author Response · Authors · 2025-11-22
> **Response to Jigsaw-R1 Authors**
>
> We thank the authors of Jigsaw-R1 for the encouraging comment and for bringing attention to the connection between our works. We find Jigsaw-R1 to be an inspiring precursor in the exploration of rule-based visual RL for MLLM post-training, and we have discussed it in our related work.
>
> Regarding the experimental comparison, we are very interested in discussing a significant divergence we observed regarding the "solvability" of the jigsaw task.
>
> In Jigsaw-R1, we noticed that the model appears to struggle with relatively simple **2x2 image jigsaws**, achieving lower accuracy (around 10-20%). In contrast, our experiments demonstrate that, with our training setup, the model can achieve approximately 70% accuracy even on more complex 3x3 jigsaws.
>
> We are very curious to hear your insights on the potential reasons for this performance gap. We raise this point because our ablation studies suggest that the difficulty level and the model's mastery of the jigsaw task are directly correlated with the magnitude of improvement on downstream tasks.
>
> We look forward to your thoughts!

---

> > ### Public Comment · ~Junyi_Zhu1 · 2025-11-23
> > **Comment by Jigsaw-R1 Authors**
> >
> > We appreciate you looking closely at our results! We would like to offer a quick clarification regarding the 2x2 performance figures.
> >
> > The lower accuracy (10–20%) you referenced corresponds to an **out-of-distribution** setting in our paper, where the model was trained solely on 2×1 grids but evaluated on 2×2 puzzles. When trained and evaluated on the same configuration (2×2), our method actually achieves **near-perfect accuracy (~100%)**.
> >
> > We chose not to scale up to puzzles larger than 2×2 because our experiments (specifically with Qwen2.5-VL-3B) indicated that smaller puzzles (2×1 and 3×1) **yielded superior downstream generalization**. This suggests an interesting trade-off where harder pretext tasks do not necessarily guarantee better transfer, particularly for models at the 3B scale.
> >
> > Given this, we are curious: how does your method compare to ours in the 2D setting in terms of downstream generalization?

---

> > > ### Author Response · Authors · 2025-11-27
> > > **Response to Jigsaw-R1 Authors**
> > >
> > > Thanks for the clarifications. In Figure 3 of our paper, we demonstrate that the 2*2 image jigsaw also brings improvement on downstream benchmarks, but unlike the conclusion from Jigsaw-R1, we find that a harder configuration (3*3) further increases the performance.

---

### Author Response · Authors · 2025-11-18
**General Response**

We thank all four reviewers for their careful and constructive feedback. The reviewers generally agree that our work presents an elegant, simple yet effective post-training framework (Dhxc,znHd,uvM4). They highlight that the paper is well written, easy to follow (Dhxc,uvM4,kYxc), and the method achieves strong and consistent gains across image, video, and 3D benchmarks (Dhxc,uvM4).

One main concern (Dhxc,uvM4,kYxc) is that qualitative analysis of the thinking traces and the analysis of the performance improvement are missing. In the response and the revised manuscript, we have added and analyzed qualitative reasoning traces,  and provided a “what is learned” analysis tying sub-skills to specific benchmarks. Another concern is about experiments on more base models (Dhxc, uvM4), harder configurations (Dhxc, znHd), and possible task combination (kYxc). We have conducted new experiments to show the general effectiveness of our method.

There are also some comments about some under-emphasized related methods and improper claims when discussing the related work (Dhxc, kYxc). We have added corresponding discussions and comparisons and revised our claims in the updated paper.

We hope these revisions address the main concerns and kindly ask the reviewers to re-evaluate the paper based on the response and the updated version.

---

### Author Response · Authors · 2025-11-18
**Paper Revision**

We appreciate the reviewers’ suggestions, which helped us to refine our manuscript. We have modified our submission based on the feedback from the reviewers and summarized the main revised points below:

* **Analysis and qualitative examples.** We have added representative `<think>...</think>` traces for all three modalities and discuss the patterns in Section 4.3 and Appendix A.1.

* **Related work & positioning.** We expanded the discussion of reconstruction-based MLLM methods, modern jigsaw-style SSL, and o3-like visual RL / “thinking with images” approaches, and softened claims that previously understated these lines of work in the Related Work section.

* **Training cost and RL vs. SFT.** We added detailed information on training cost for all three modalities and provide a comparison between SFT and RL in Sections 4.1 & 4.3.

* **New experiments** We include new experiments with harder jigsaw configurations and a new base model. The results are added in Section 4.3.

In the updated manuscript, we have marked the main changes and newly added material in blue. If there are any remaining questions or suggestions, we are very happy to address them during the discussion phase and will incorporate further feedback into the final version.

---

### Author Response · Authors · 2025-12-03
**Summary for AC Consideration**

Dear Area Chair,

We sincerely appreciate your time and effort in handling our submission. We have actively engaged with all reviewers to improve the manuscript. Below is a summary of our rebuttal efforts and the current status of the paper.

**Acknowledged Strengths**
Reviewers consistently appreciated the following aspects of our work:
*   **Elegant & Novel Framework:** The Visual Jigsaw task is recognized as a simple, effective, and architecture-agnostic post-training framework (Dhxc, znHd, uvM4).
*   **Versatility:** The method proves effective across three distinct modalities: Image, Video, and 3D (Dhxc, uvM4).
*   **Clarity:** The paper is well-structured, easy to follow, and the motivation is clear (Dhxc, uvM4, kYxc).

**Rebuttal Efforts Summary**
We have addressed the major concerns raised by reviewers through extensive new experiments and analyses:
*   **Mechanism & Qualitative Analysis:** We added representative `<think>...</think>` traces and a "what is learned" analysis to demonstrate how the model develops specific sub-skills (Addressing Dhxc,uvM4,kYxc).
*   **Generalization & Multi-Tasking:** We conducted new experiments showing our method works on a different base model (MiMo-VL) and succeeds in a multi-task setting (joint Image + 3D training), as well as demonstrating cross-domain transfer capabilities (Addressing Dhxc, uvM4, kYxc).
*   **Harder Configurations & Cost:** We explored harder puzzle settings (4x4 grids, 8-clip videos) with detailed failure/success analysis and provided a full training cost breakdown comparing SFT vs. RL (Addressing Dhxc, znHd).
*   **Related Work & Claims:** We refined our claims regarding text-centric paradigms and added discussions on reconstruction-based methods and recent Jigsaw-based SSL works (Addressing Dhxc, kYxc).

**Current Status**
*   **Reviewer kYxc** has explicitly acknowledged that their concerns regarding claims were addressed and appreciated the new multi-task experiments. In a second round of discussion, we further addressed their follow-up questions by demonstrating successful cross-domain transfer (Image + 3D) and verifying that our method remains effective in "thinking" mode using the MiMo-VL baseline.

We believe our revisions have significantly strengthened the paper’s contribution to self-supervised MLLM post-training.

Best regards,
The Authors

---

### Meta-Review · Area_Chair_UFaJ · 2025-12-28

**Summary:**

This submission was reviewed by 4 knowledgeable reviewers, who raised concerns about:
1. The novelty of the approach, which appeared incremental (Dhxc)
2. Missing discussion and positioning w.r.t. some prior art (Dhxc) and validity of some of the claims (kYxc)
3. The method's training overhead (Dhxc)
4. The evaluation of the proposed approach:
    - The method was initially tested on one single model (Qwen2.5-V), leaving its generalizability unvalidated (Dhxc, uvM4)
    - Missing qualitative analyses to substantiate some claims and showcase how the proposed supervision may enhance visual understanding (Dhxc, uvM4, kYxc)
    -  Missing comparisons to SoTA for image and video benchmarks (Dhxc)
    - Unclear significance of some improvements (znHd)
    - Unclear rationale behind the choice data to perform visual jigsaw post-training (uvM4)

**Reviewer Concerns:**

During rebuttal and discussion, the authors addressed the reviewers' concerns. In particular, the rebuttal argued that the core innovation of the work lies in demonstrating how vision-centric SSL task may be adapted for MLLM post-training. The rebuttal also positioned the work w.r.t. missing references, clarified the validity of the claims made, and discussed the additional computation required, concluding that the overhead may be justified by the improvements achieved.

The rebuttal improved the evaluation of the proposed approach by testing the framework on MiMo-VL-7B-SFT-2508. Given the kind of task introduced in the paper (jigsaw pretext tasks), it would have been interesting to assess the impact of the tasks when used beyond post-training (e.g., pretraining or prior to instruction-tuning). The rebuttal also shared qualitative analyses, and addressed the significance of improvements questions by reporting std for a subset of results.

Finally, the rebuttal clarified that the visual data used for post-training comes from widely-used benchmarks oftentimes used for instruction-tuning.

**Reviewer Scores:**

Given that the main concerns raised by the reviewers (novelty, experimental validation) were addressed during rebuttal/discussion, I would have expected the reviewers who gave 4 to slightly increase their scores. Therefore, the AC recommends to accept.

---

### Decision · Program_Chairs · 2026-01-26

Accept (Poster)